# Validation of a spring loaded probe for single and repeat pressure pain testing, including public domain specifications for design and manufacture

**Søren O'Neill**[1,2]*, **Casper Glissmann Nim**[1,2,3], **Natalie Hong Siu Chang**[1]

1 Medical Spinal Research Unit, Spine Centre of Southern Denmark, University Hospital of Southern Denmark, Middelfart, Denmark, 2 Department of Regional Health Research, University of Southern Denmark, Odense, Denmark, 3 Center for Muscle and Joint Health, Department of Sports Science and Clinical Biomechanics, University of Southern Denmark, Odense, Denmark

* soeren.oneill@rsyd.dk

**Data Availability Statement:** All relevant data including raw data are within the paper and its Supporting Information files.

## Abstract

Temporal summation of pressure pain is technically more challenging than simple pressure pain thresholds. The current study describes the design, manufacture and validation of a simple mechanical test apparatus to assess the temporal summation of deep pressure pain. We release design details into the public domain with the intention of providing free access for researchers especially in low income countries. Utility and validity of the probes were assessed by pressure application in three different experimental setups: A. Identifying potential issues which needed to be addressed to ensure a reliable test procedure (189 tests with 24 testers using four different probes). B. Selecting the most reliable target force curve (one tester conducted 20 tests). C. Estimating classic inter and intra-examiner reliability and comparing probe measures to other QST measures (repeated measures study with counterbalancing). We make recommendations on best use of the probes. Pressure pain thresholds assessed using probes were affected by anatomical test site and testing tool, but not by tester, day or session. Temporal summation of pressure pain was significantly greater than that of a single pressure application. We found no correlation between temporal summation using the probes on the Infra-Spinatus muscle and temporal summation using a pneumatic cuff on the lower leg. The probe was a useful tool for assessing pain intensity and temporal summation of pressure pain intensity, but not for pain thresholds. A number of caveats need to be considered when using the probe, including but not limited to audio cues and target ideal wave function.

## Introduction

The literature on *temporal summation* (TS) of pressure pain and its relation to clinical status is not entirely clear, but there is evidence e.g. that experimental pain sensitivity including temporal summation of pain is predictive of clinical aspects such as fear-related activity avoidance

**Funding:** This work is part of a large study on pain sensitivity in chronic low back pain, which received funding from the Danish Foundation for Advancement of Chiropractic Research and Postgraduate Education. Beneficiary of the grant: SON Grant number: A1298 Full name of funder: Foundation for Advancement of Chiropractic Research and Postgraduate Education Funder URL: https://rltn.dk/fonde/praksisfondene/fond-til-fremme-af-kiropraktisk-forskning-og-postgraduat-efteruddannelse The funders had no role in study design, data collection and analysis, decision to publish, or preparation of the manuscript.

[1]. In a recent review on *low back pain* (LBP), den Bandt et al. [2] concluded that tests of temporal summation had mixed outcomes. However, the publications included in that review relied primarily on superficial pin prick rather than pressure stimulation of deeper musculoskeletal structures. There is some evidence that temporal summation of deep pressure pain is increased in chronic musculoskeletal pain states such as LBP [3] and osteoarthrosis of the knee joint [4].

As TS relies on central integration of repeated stimuli, it may prove more relevant to central sensitization than simpler *quantitative sensory tests* (QST) such as pressure pain thresholds. The effect of temporal summation has been found to be more pronounced with stimulation of deep structures than superficial ones [5] and a greater TS effect is observed with shorter inter-stimulus intervals, than longer intervals [6], which suggests that the specifics of TS testing paradigms may be important. The TS effect is also sensitive to other factors affecting pain sensitivity, such as delayed onset muscle soreness [7].

Assessing TS of pressure pain is technically more challenging than simple pressure pain thresholds as the pressure needs to be applied repeatedly and consistently with a controlled force and rate of application. This will prove difficult to achieve using a hand-held algometer–especially so if the inter-stimulus-interval is to be kept low, as suggested by Lautenbacher et al. and Nie et al. [5, 8].

To assess deep mechanical pressure pain with repeated and consistent pressure applications, Graven-Nielsen, Mense and Arendt-Nielsen reported on a method based on a computer-controlled linear actuator, which was mounted in a frame over the test bed [9]. Lautenbacher et al and Nie et al used a similar method [5, 8]. Temporal summation of deep tissue pain has also been tested using cuff algometry, in which a cuff is secured around a limb and inflated to induce mechanical pressure [10]. Although significantly less troublesome to install and operate than the linear actuator setup, cuff algometry has yet to gain widespread clinical use, perhaps due to high costs. Also, its application is restricted to the limbs.

In summary, temporal summation of deep mechanical pressure may provide clinically meaningful information about pain sensitivity, but a clinically applicable QST procedure is not readily available.

The current study was conducted to design, manufacture and validate a simple '*bed-side*' mechanical testing tool to assess temporal summation of deep pressure pain.

## Methods and materials

### Design of the pressure probe prototype

After constructing and experimenting with a number of hand-made prototypes, we concluded that it would be impractical to produce a single pressure probe capable of delivering an adjustable yet adequately specific force within a sufficiently wide range. We found the most practical and efficient construction to be a single universal probe platform, which could be adapted with different springs to provide a fixed specific pressure, within the range of 1–10 kg.

This was achieved by constructing a probe with an internal mechanism which would accommodate different mechanical springs that met certain technical criteria. For each probe, the required force could thus be approximated by selecting a spring with the appropriate characteristics. The probe force would be fine-tuned by carefully calibrating the spring pre-load using the internal adjustment mechanism.

A photo of the final probe version is presented in Fig 1 and a schematic exploded view of internal components and nomenclature is presented in Fig 2.

After procuring springs with appropriate technical characteristics we constructed four probes corresponding to 2kg, 4kg, 6kg and 8kg of force at approximately 2cm compression. E.

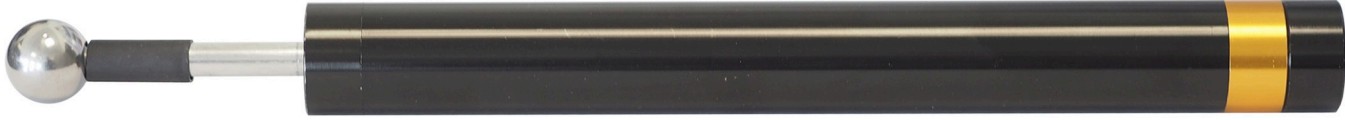

**Fig 1. Pressure pain probe.** Photo of a mechanical pressure probe, suitable for repeated pressure applications.

g. to achieve a probe pressure of 2kg we procured a spring with a spring-constant of 1.02 N/mm. The spring in question had a suitable loaded outer diameter and unloaded resting length to be housed within the probe handle.

In earlier experimentation with handmade prototypes, we had learned that it was necessary to ensure sufficient distance between the pressure probe head and the front end of the probe handle when in the compressed position. Otherwise there was a risk of pinching the test subjects skin between the handle and the probe head. To avoid pinching the skin, two things were needed: a) that the desired probe force was reached at a degree of spring compression which left a sufficient safety distance between probe handle and head to avoid pinching, and b) that an indicator alerted the tester when the correct degree of spring compression was reached. Regarding a), this was easily solved by using a probe push rod of sufficient length, see Fig 2. Regarding b), this required an indicator on the probe push rod to alert the user when the correct compression of circa 2cm had been reached. We experimented with superficial Sharpie pen markings, with small deformations in the metal surface of the push rod, and an adjustable metal collar on the push rod. In the end, using a 3D-printed hard plastic tubing of the right diameter proved both simple and efficient. It allowed for easy re-adjustment and importantly provided a tactile rather than visual clue to the tester that the correct spring compression had been reached and further pressure application would overshoot the target force.

**Fine-tuning.** As the probes needed to be calibrated, a calibration bench was required which could provide a measurement of the force exerted by the probes when in the compressed state, see Fig 3. Arguably, this could have been achieved using a simple, commercially available scale or even fixed weights, but the calibration platform as described here allows for accurate (+/- cirka 0.005kg) time-series data capture on a computer.

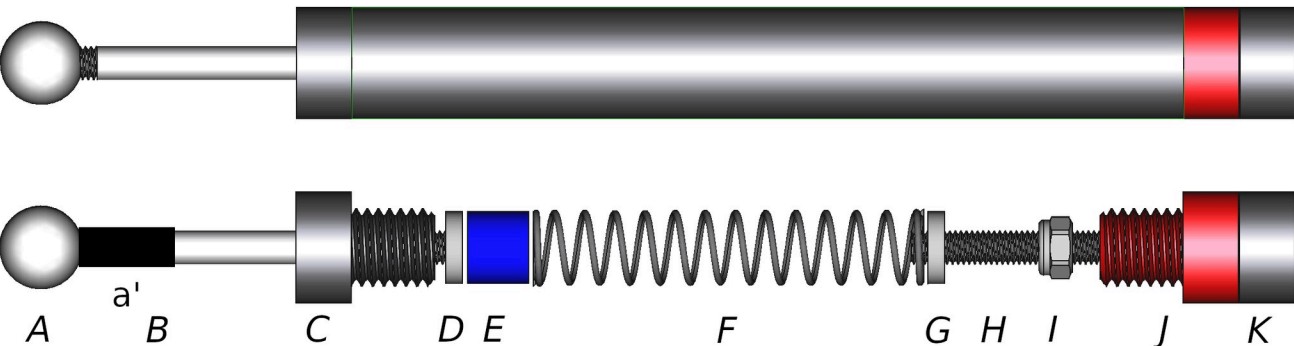

**Fig 2. Technical schematic.** A schematic of the probe with the outer tube removed and the internal mechanism exposed. The probe consists of a spherical pressure probe head (A), a section of plastic tubing (a'), a push rod (B), a front tube end (C), a front spring push plate (D), a spring spacer (E), a spring (F), an outer tube (not shown), a rear spring push plate (G), a threaded spring preload adjustment rod (H), an optional retaining nut (I), a rear tube end (J) and a retaining nut (K). The head (A), push rod (B) and front push plate (D) are joined together and move as one, back and forth in the smooth internal bore of the front tube end (C). The threaded adjustment rod (H) and rear push plate (G) are joined togther and their position can be adjusted back and forth in the internal threaded bore of the rear tube end (J). By counter-tightening the rear tube end (J) and the retaining nut (K) the threaded adjustment rod (H) can be fixed in place, thus setting the spring (F) preload. A spring spacer (E) may be required to take up any slack, depending on the length of the spring (F) in use.

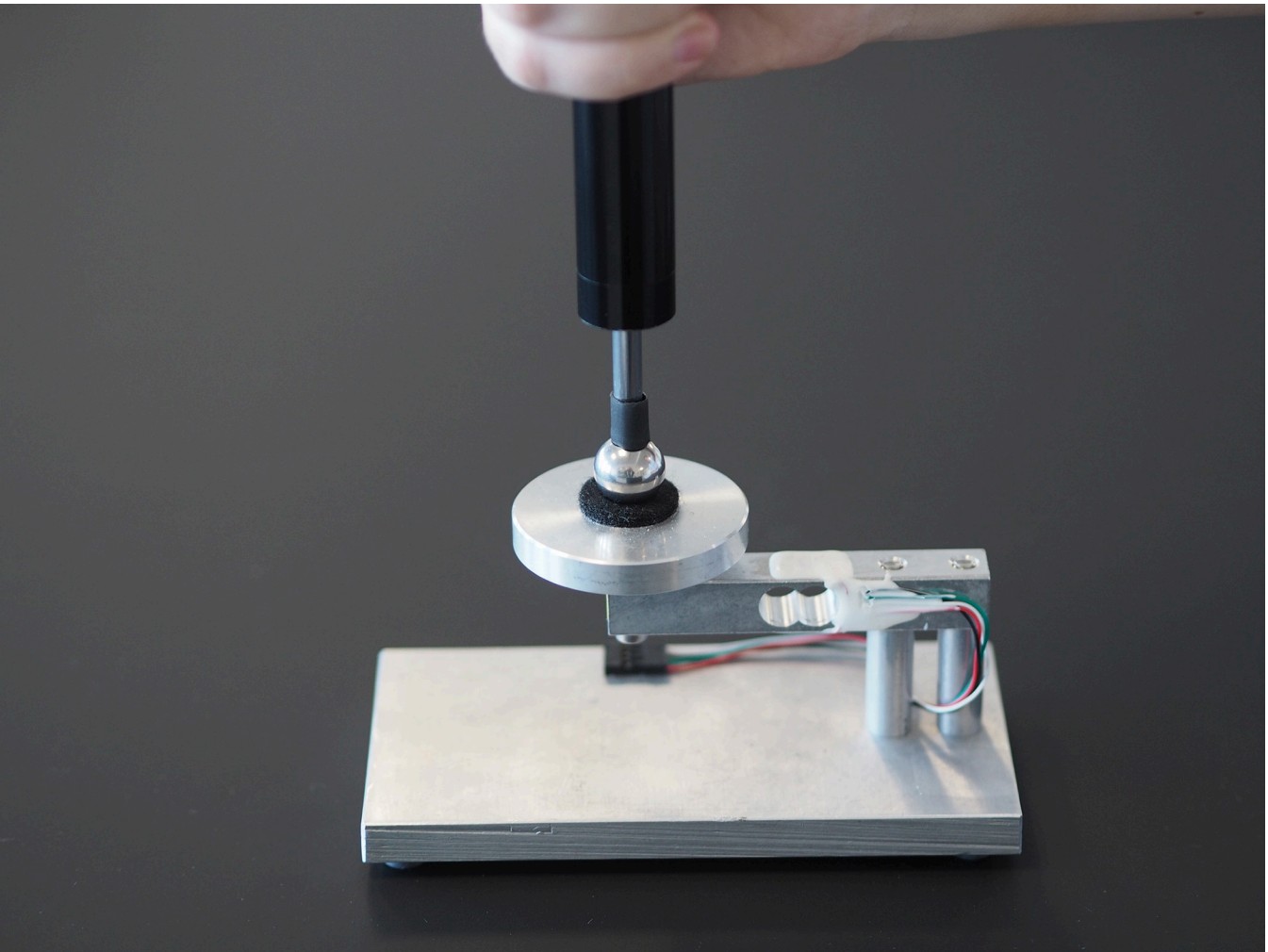

**Fig 3. Calibration tool.** Testing and calibration platform used to assess force development during repeated pressure application.

To prevent the pressure probes from slipping, a self-adhesive felt pad served as the contact point for the probes on the platform.

The test platform consisted of an aluminum parallel beam load cell with an HX711 amplifier and an analog-to-digital converter. The digital signal was processed using an Arduino Nano v3 (ATmega328 processor based circuit board–see https://store.arduino.cc/collections/boards/products/arduino-nano for details) and serially transmitted (9600 baud) to a Linux PC (Ubuntu 22.04) from which the raw serial data stream could be read and analyzed.

The digital signal was passed to a custom coded computer application (Python3, GTK3) which continuously recorded, displayed and stored the load cell sensor data.

To ensure accurate calibration of the pressure probes, the mechanical calibration-bench was in turn calibrated using fixed weights obtained from an accredited commercial enterprise specializing in weight calibration (Kern & Sohn GmbH, Ziegelei 1, 72336 Balingen—Germany).

**Probe head.** In early experimentation, we used a pressure probe head similar to conventional algometers, i.e. a flat circular contact surface of 1cm$^2$ with rounded edges. However, with repeated application of pressure in the same position as required for temporal summation testing paradigms, the mechanical strain on the skin at the border of the probe head was

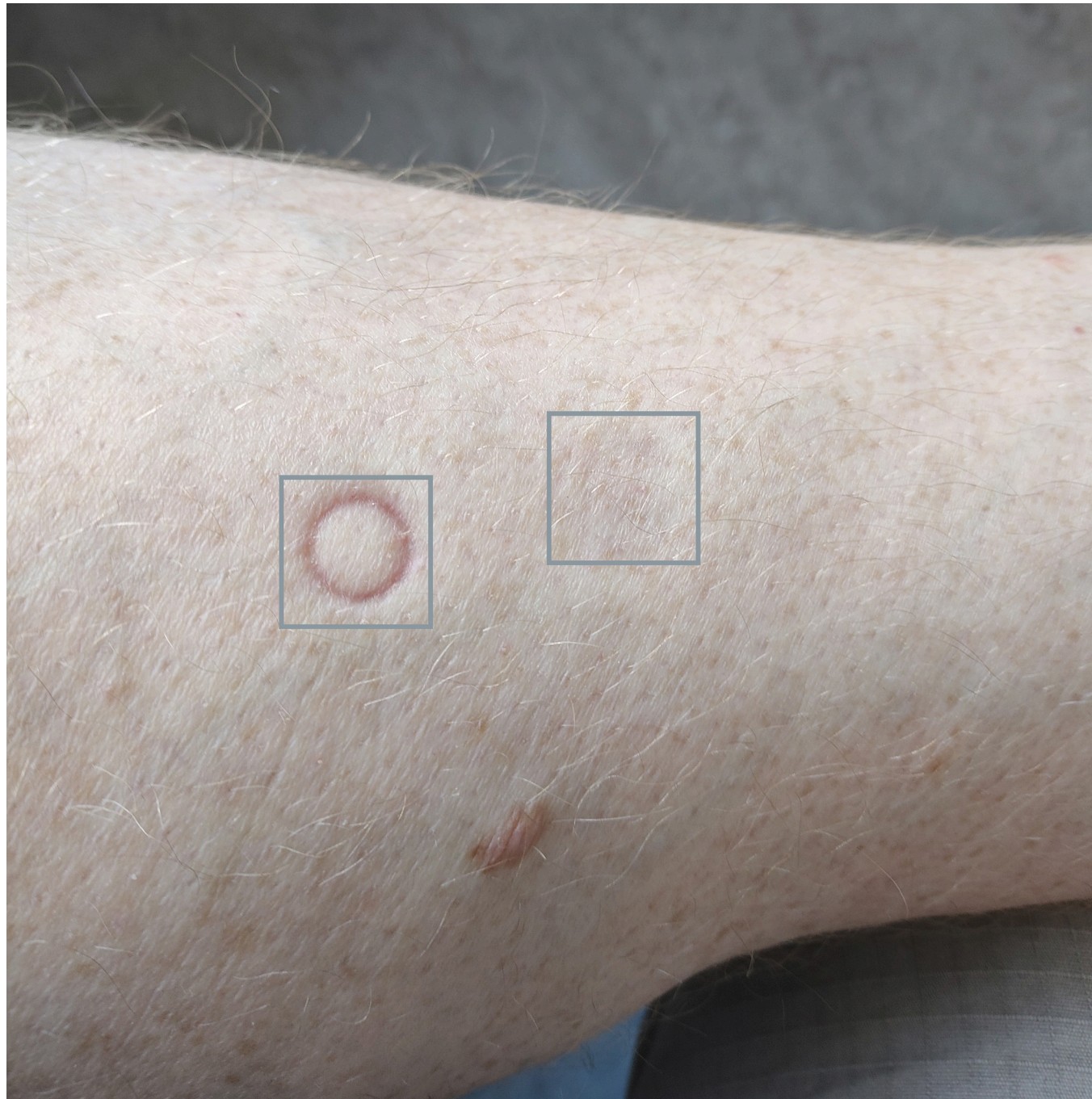

**Fig 4. Skin marks.** Two minutes after 10 repeated applications of 8kg in 1 s intervals using two different probe heads. A flat circular probe head with rounded edges (left box) leaves clear marks as indication of mechanical strain of the skin. A spherical probe head (right box) leaves no such markings. This is an indication that the spherical probe head is associated with less superficial strain and by extension less potential for superficial nociceptive stimulation compared to the flat probe head.

sufficient to leave red circular markings (see Fig 4). In line with previous recommendations [11], we therefore opted to use a spherical probe head instead to help ensure that pain was induced by pressure in deep somatic structures rather than stretching and potentially straining of superficial skin. The spherical probe head had a diameter of 15mm.

On the basis of what we had learnt from experimentation with early prototypes, we proceeded to test the characteristics of the pressure probe.

## Pressure probe test setup

We tested the pressure probes in three different experimental setups of which two were pre-planned and one was added post hoc:

- "Force measurement" recorded force as a function of time during repeated pressure applications on a force-tranducer platform.

- "Sine or square wave form" was added post-hoc to examine the fidelity of recorded force on a force-transducer platform, when aiming to approximate one of two different target functions (sine wave vs square wave).

- "Pain measurements" recorded pain reports as a function of force application on human volunteers.

**Force measurement.** Volunteer university students (MSc. Clinical Biomechanics, University of Southern Denmark) performed tests of repeated pressure applications. The testers used four different pressure probes color-coded as black, red, gold and silver, in two different test protocols, *Guided* and *Unguided*. Participants were invited to perform a minimum of 8 tests but were allowed to perform more. Each of these tests consisted of 10 repeated pressure applications to the testing platform shown in Fig 3.

*Participant instructions.* Initial instructions were provided as a pre-recorded video in which the entire test procedure was described accompanied by illustrative video and screen-casts of the test procedure. During the testing and data collection itself, simple on-screen instructions were provided in real time to guide the participants through the procedure: which probe to use, when to start, etc. In this way participants were left alone to perform the tests without supervision.

Participants had no prior experience using the pressure probes or the test setup and no time was allocated for rehearsal prior to testing. These were deliberate choices intended to reveal potential problem areas when instructing testers in the procedure.

*Hardware and software for testing.* The data collected by the test platform described above (Fig 3) was passed to a custom coded computer application (Python3, GTK3) which served three purposes: to continuously record and store the load cell sensor data (i.e. force applied), to randomize test order and to direct participants on how to perform the test procedure in the correct manner and order.

The test procedure consisted of a series of no less than 8 different tests: the first 4 tests were performed unguided with each of the four color-coded probes in random order. Similarly, the last 4 tests were guided using the same four probes in a different random order. In the guided protocol the custom computer application provided an audible metronome beat (sound cue) with two different click sounds alternating every 1 s to indicate when to apply and release probe pressure. In the unguided protocol no such sound cues were provided and participants were required to set the pressure application-and-release cadence unaided.

Prior to testing, the testing platform was calibrated using fixed calibration weights of 0 kg, 1 kg and 5 kg (Kern & Sohn GmbH, Ziegelei 1, 72336 Balingen—Germany). The testing platform was found to be accurate to within plus/minus 5 g.

As the probe calibration functionality described in the preceding sections was still being developed we were not able to accurately calibrate the four pressure probes to a specific target force. Instead the actual force exerted by the probes in the compressed state of 2 cm was measured. The pressure exerted by the four probes (red = 2.1 kg, silver = 4.1 kg, gold = 5.8 kg and

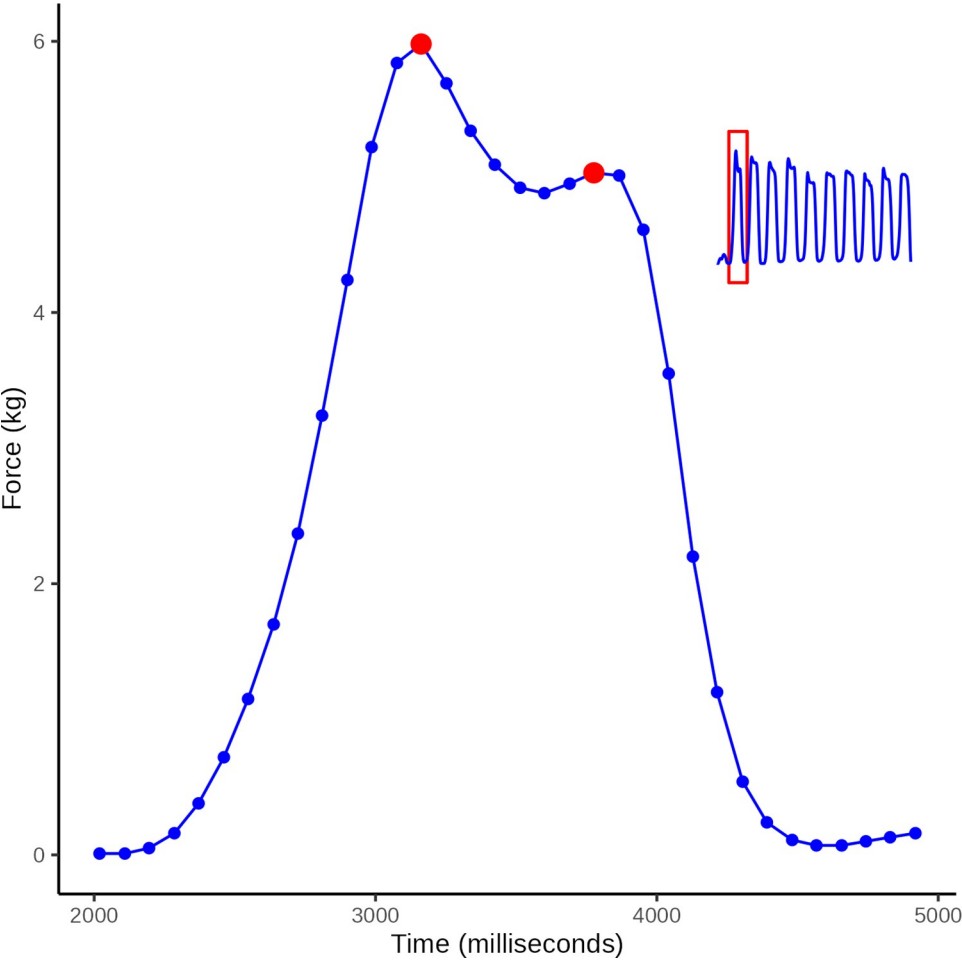

**Fig 5. Curve peak.** Illustrative plot of raw data from a single pressure application (time vs force) from a series of ten force applications (see insert). Two peaks (red) are apparent within a 1 second time frame. Of these, the one of greater value was defined as the wave peak.

black = 7.4 kg respectively) were used to define the ideal sine (sinosoidal) wave function for analyses of time series data.

*Time series data analyses.* An illustrative bi-plot of the raw data from a single pressure application is presented in Fig 5.

The observed time series data were compared to an ideal time series: A sine wave of the appropriate amplitude (2.1kg, 4.1kg, 5.8kg or 7.4kg) and a period of 2 s.

The observed time series data were sampled at the highest possible sampling rate permitted by the test equipment (Arduino platform, weighing cell and A-D converter), which was found to be a mean of 87.78 ms between sensor readings (SD = 2.46 ms). The ideal sine wave time series was thus sampled with a corresponding sample rate of 87.78 ms.

Using the guided protocol with a sound cue, the time dimension and the number of samples could thus be assumed to be relatively comparable between the observed data and the ideal sine wave data. For the unguided protocol however, the time dimension and consequently the number of samples could not be assumed to be comparable. This represented a problem for our wave signal analyses, as the number of data points could not be assumed to be similar between curves.

Prior to analysis of the observed and ideal time series, the data were therefore manipulated using *dynamic time warping* (DTW), an algorithm suitable for measuring similarity between

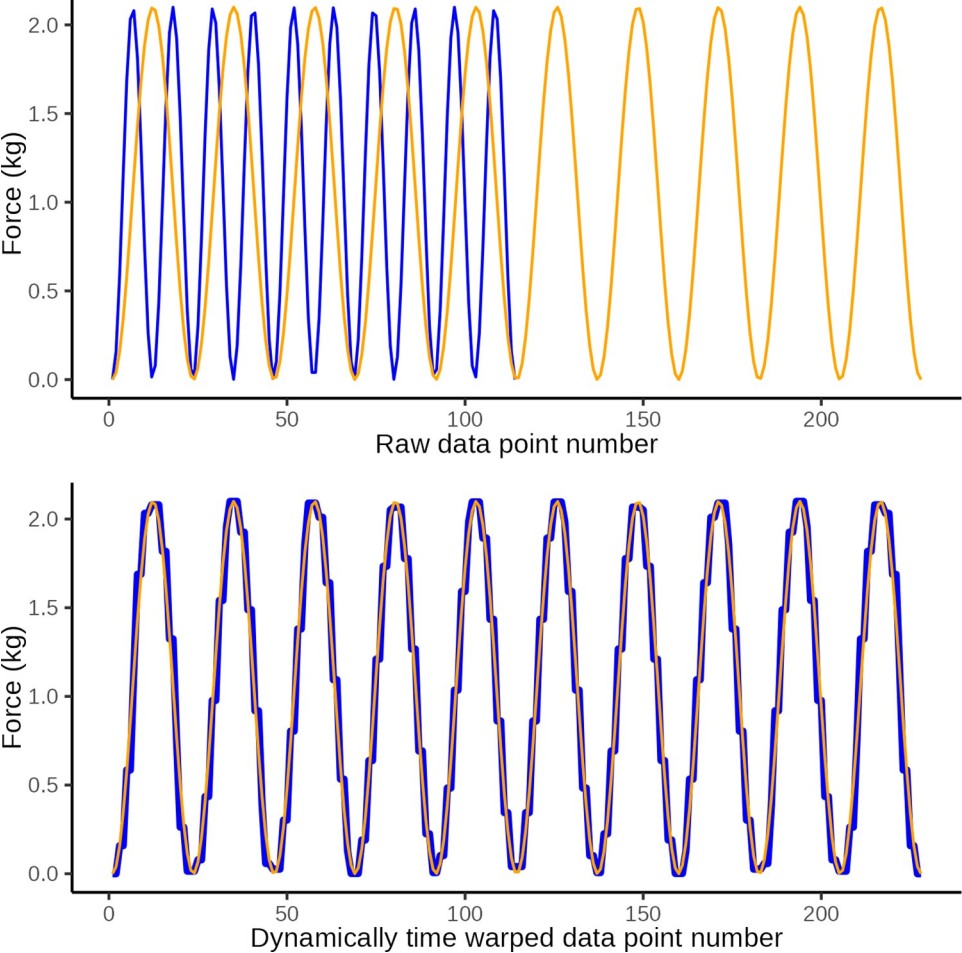

**Fig 6. DTW illustration 1.** Illustrative plot of constructed observational data (blue) in which the wave period is too short at only 1 s, and the ideal target sine wave (orange) with a wave period of 2 s The top graph illustrates the original data and the lower graph illustrates the same data after dynamic time warping (y-axis: force in kg, x-axis: observation number corresponding to time). Dynamic time warping optimally *maps* the 114 data points in the observed curved with the 228 data points in the ideal curve. A Eucledian distance of 20.3 quantifies the disparity between curves in the lower graph which is primarily ascribed to disparity introduced by differences in number of data points (i.e. jagged curved caused by temporal disparity).

two temporal time series in which phases, number of samples, and length may vary. DTW is commonly used in analyses of speech audio signals and similar time-series signals [12].

In DTW the observed time series is "warped" non-linearly in the time dimension only, to allow for a measure of similarity between curves when they are optimally synchronized in time. DTW is done by matching data points on a two-way one-to-many basis so that the time series curves with the same patterns are optimally matched. To illustrate this we present three constructed examples in Figs 6–8. These illustrate a simulated series of observational data (blue) and an ideal sine wave curve (red) for comparison, before and after DTW.

After DTW we calculated the Euclidean distance as a measure of the similarity between the observed data and the ideal sine curve. This is a measure of the vertical disparity between matched data points between the two curves of which one is time-warped. The smaller the Euclidean distance is (it is always $\geq 0$), the better the performance, i.e the closer the observed time series was to the ideal sine curve, after being dynamically warped in the time axis.

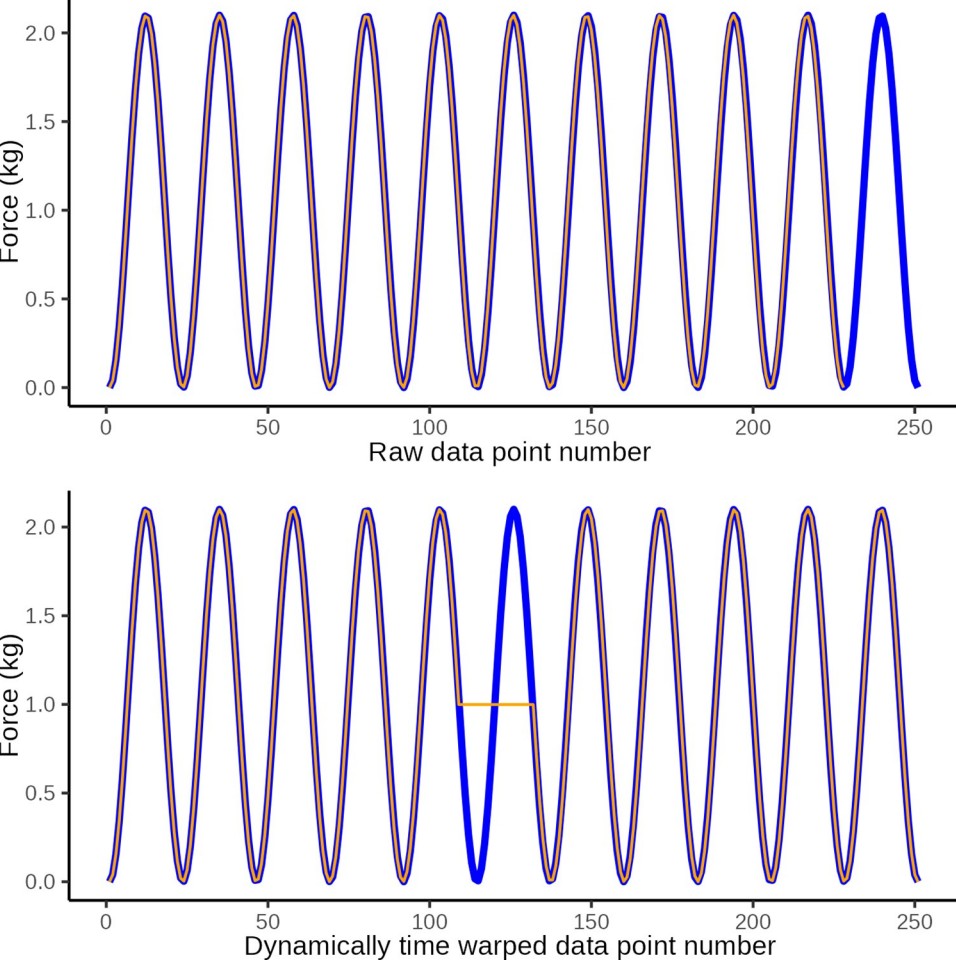

**Fig 7. DTW illustration 2.** Illustrative plot of constructed observed data (blue) with an extranumary pressure application (wave) and the ideal target sine wave (orange). The top graph illustrates the original data and the lower graph illustrates the same data after dynamic time warping (y-axis: force in kg, x-axis: observation number corresponding to time). Dynamic time warping optimally *maps* the 251 data points in the observed curved with the 228 data points in the ideal curve. A Eucledian distance of 18.1 represents the disparity between curves in the lower graph which is primarily ascribed to disparity introduced by the extranumery data points evident at the 6th sine wave.

In essence, the Euclidean distance is thus a measure of the similarity of two time series of data once they have been warped non-linearly along the time dimension to ensure that the two curves are optimally in step. The better the tester can approximate the shape of the ideal curve, the lower the Euclidean distance.

Three conditions (boundary, monotonicity, and continuity constraints) must be met in order to comply with the restrictions of the Warping Function as described by Sakoe and Chiba [13], such that head and tail must be positionally matched, with no cross-match and no left-out observations.

The optimal match is denoted by the match that satisfies all the restrictions and that has the optimal, lowest cost, which is computed as the sum of absolute differences, for each matched pair of data-points. Two time series, X and Y (e.g. a measured signal and an ideal signal), can thus be arranged to form an n-by-m grid, where each point (ix, iy) is the alignment between x [ix] and y[iy].

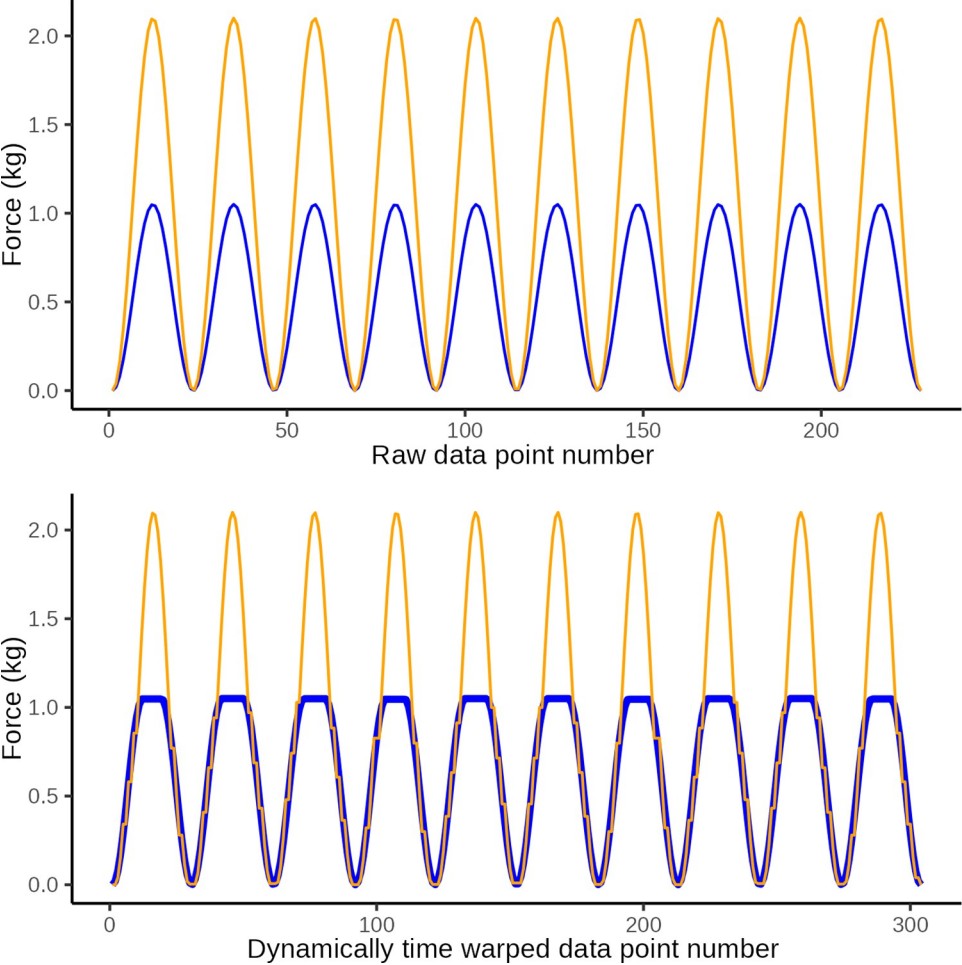

**Fig 8. DTW illustration 3.** Illustrative plot of constructed observed data (blue) with low amplitude of pressure applications and the ideal target sine wave (orange). The top graph illustrates the original data and the lower graph illustrates the same data after dynamic time warping (y-axis: force in kg, x-axis: observation number corresponding to time). Dynamic time warping optimally *maps* the 228 data points in the observed curved with the 228 data points in the ideal curve. A Euclidian distance of 86.0 represents the disparity between curves in the lower graph which is primarily ascribed to disparity introduced by different amplitudes.

The signals were aligned using DTW, and the minimum Euclidean distances were calculated by finding the root sum of squared differences (see Eq 1):

$$d_{mn}(X, Y) = \sqrt{\sum_{k=1}^{K} (x_{k,m} - y_{k,n}) * (x_{k,m} - y_{k,n})} \tag{1}$$

Eq 2 describes the way to obtain the optimal warping path by minimization of the Euclidean distance.

$$d = \sum_{\substack{m \in ix \\ n \in iy}} d_{mn}(X, Y) \tag{2}$$

For further details see MATLAB at https://se.mathworks.com/help/signal/ref/dtw.html.

The sum of Euclidean distances are thus an expression of the overall disparity between the observed time data and the ideal sine wave function. In other words, how similar are the curves after DTW has minimized the effects of time-disparity.

Prior to using DTW, the observed time series data were inspected for signal errors and outliers. Also, each observed data time series was aligned on the x-axis with the ideal sine wave to ensure that the very first peak of the measured time series was aligned with the very first peak of the corresponding ideal sine wave. This eliminated any curve-disparity introduced by participating testers pausing before starting the test protocol.

*Summary data analyses.* Data from each data series were also summarized as **a)** specific number of pressure applications (target = 10), **b)** mean time lapse (wave period) between pressure peaks (target = 2 s) and **c)** mean peak pressure (target = 2.1 kg, 4.1 kg, 5.8 kg, and 7.4 kg respectively).

Visual inspection of plots of observed time series data, revealed that pressure application cycles were easily identified. The peak force of each wave was determined as the highest value observed within a 1 s time frame. For an illustration, see data points highlighted in red in Fig 5.

## Sine or square wave form

As described above, the guided protocol included a sound cue consisting of two different 'tick' and 'tock' sounds indicating when to apply and release pressure, respectively. Also, testers were instructed to apply the target pressure, maintain pressure for 1 s, release it and wait for 1 s before re-applying pressure. This was the case for both the guided and unguided protocols.

Although not explicitly stated, this suggests that the ideal time-force function was that of a square wave. With its instantaneous increases and drops in pressure, it is obviously only possible for a human tester to approximate a square wave function. Furthermore, as a human tester attempts to approach the abrupt increase in force of a square wave there is a greater risk of overshooting and applying too much pressure with each application.

This was indeed observed to be the case in the present data, and thus an additional smaller post-hoc test was performed to compare force curves approximating two different ideal time-force curves: a square wave or a sine wave.

To examine and contrast these two strategies, a single tester (SON) conducted 2 sets of 10 tests of 10 pressure applications using a 4.0 kg probe. The applied force was measured using the platform illustrated in Fig 3. The 2 sets of 10 tests were guided by 2 different sound cues: One consisting of the same tick/tock sound as that described above in the assessment of time-force functions. The other sound cue was a constant tone which fluctuated in volume as a sine function. After a few practice sessions, the tester performed 10 tests (of ten applications) in short succession in which the tester attempted to apply force as a square or sine wave respectively.

Data was handled as described above with DTW and calculation of Euclidean distances, but in relation to the two different ideal curves: 10 square waves or 10 sine waves.

## Pain measurements on human volunteers

Reliability of the pain induced by the pressure probes was assessed separately as pressure pain threshold and pressure pain intensity, with single and repeated pressure applications. These tests were performed with a different probe head: A flat circular contact with an area of $1cm^2$. As described above, such a probe head places higher strain on the skin's superficial structures than a spherical one, but we chose this probe head to allow for comparison with existing research publications on pressure pain threshold and intensity.

Four testers tested volunteer participants. Two of the testers (SON and CGN) are researchers with considerable experience using quantitative sensory testing (QST) procedures and 2 were Masters students who were instructed, trained and supervised in the use of the specific

tests. The volunteer participants were all senior university students (MSc. Clinical Biomechanics, University of Southern Denmark).

See Fig 9 for an illustration of QST test sites and types.

**Pain threshold.** The pressure pain threshold was assessed first on the right calf muscle in the middle of the Gastrocnemius muscle belly and following that, in the lower back on the right hand side, 5 cm lateral to the L4 vertebra.

Pressure pain threshold was assessed using a set of nine pressure probes calibrated to exert a pressure of 2 to 10 kg in steps of 1kg. The pressure pain threshold was determined in a split middles method. Initially a pressure of 5kg was applied for 2 s. The participant was instructed to indicate whether this was perceived as painful ('yes') or not ('no'). Depending on the participants response, the next probe (3 kg if 'yes' and 8 kg if 'no') were tested. This was repeated until the lowest pressure reported as painful was determined. The probes were positioned on the skin in such a way that it only partly overlapped with previous pressure applications.

For comparison, pressure pain thresholds were also assessed using a handheld algometer in the same locations on the left-hand side and with the same probe head. Pressure was applied as a gradually increasing force of approximately 0.5 kg/s until the participant pressed an indicator button attached to the algometer, to indicate that the pressure was becoming painful.

The pressure pain threshold was determined twice for both the probes and the algometer, with at least a minute of rest between tests.

Finally, for further comparison, the pressure pain threshold of the right calf was also assessed using cuff algometry (NociTech, Copenhagen, Denmark). A cuff torniquet (13.5 cm wide) was secured around the bulk of the right calf. Pressure was increased gradually at 1 kPa/s and participants indicated the resulting pain perception using a slider on an attached physical VAS scale, ranging from 'No pain' to 'Worst pain imaginable'. The pressure pain threshold was recorded as the pressure which elicited the first VAS recording 0.

**Pain intensity and temporal summation.** Pressure pain intensity was assessed after pain threshold measurements using the 5kg probe applied to the right Infraspinatus muscle (mid muscle belly) with sustained pressure for 3 s. Participants indicated pain intensity on a digital visual analog pain scale (0–100) anchored 'No pain' and 'Worst pain imaginable'. Thereafter, 10 repeated pressure applications of approximately 1 s with 1 s rest intervals were applied in the same location. After the 10th pressure application, participants indicated pain intensity using the same VAS. Temporal summation of pressure pain was calculated as the difference in pain intensity with one 3 second pressure application and 10 repeated pressure applications.

For comparison, the pressure pain threshold of the right calf was assessed using cuff algometry based on the individual pressure pain tolerance threshold. The pain tolerance threshold was determined in the same manner as the pain (detection) threshold: Pressure was increased gradually at 1 kPa/s until the participant discontinued the test due to intolerable pain, or a maximum of 100 kPa was reached. For temporal summation, a series of 10 stimuli at the individual pain tolerance threshold was applied at 1 s intervals and pain was indicated continuously using the VAS. Temporal summation was calculated as the difference in indicated pain intensity (VAS) between the 10th and the 1st pressure application.

**Test schedule.** Participants were tested a total of 5 times on 3 different days, with 2 test sessions on day One and Two, and 1 test session on day Three. Test days One and Two were scheduled 1–3 days apart and test day Three 5–14 days after test day Two. On test days One and Two, the 2 test sessions were scheduled 1–5 hours apart.

The 4 different testers (W, X, Y and Z) were assigned into 2 groups (W+X and Y+Z). The testers tested the participants in a stratified manner such that **a)** the tester group which performed tests on day One also performed the tests on day Two, **b)** the 2 sessions on the same day were always performed by different examiners (e.g. W and X), **c)** the session-order of the

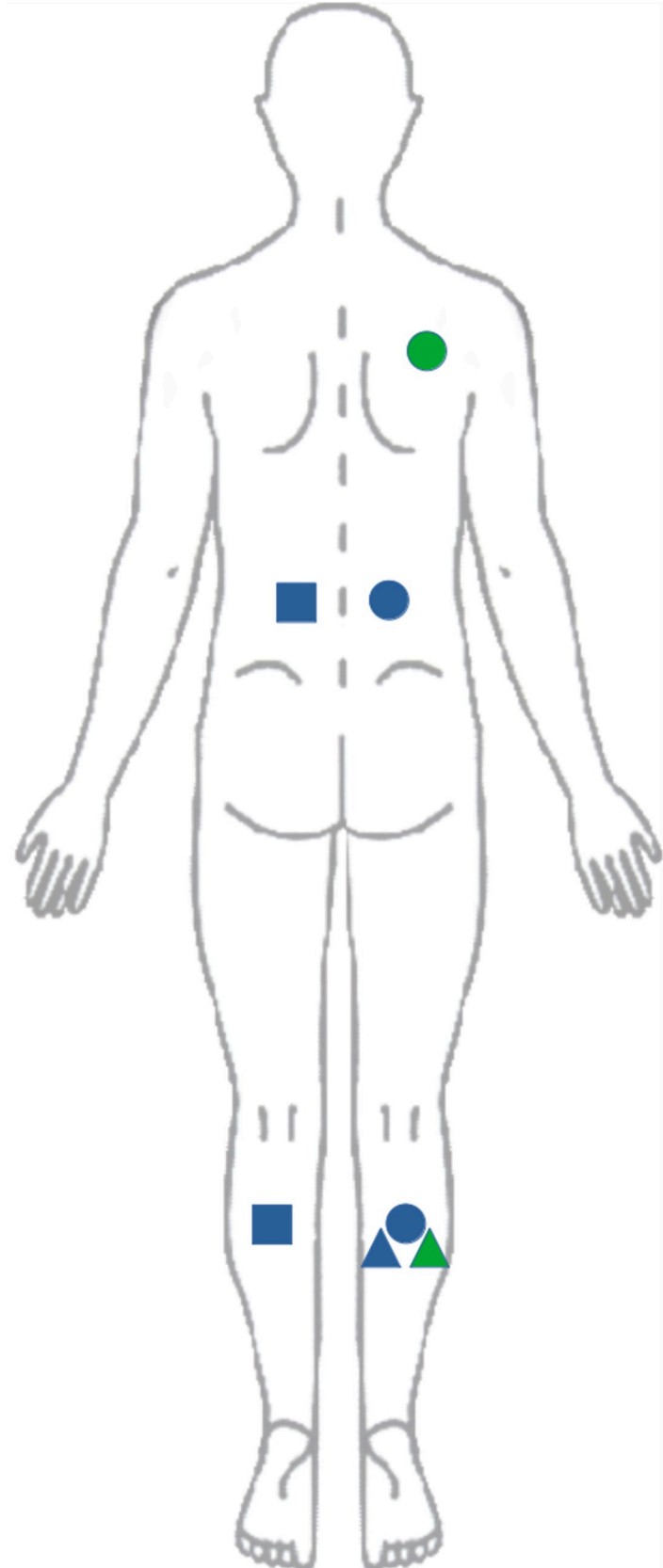

**Fig 9. Test sites.** Pain sensitivity was tested in several locations using the probes (circle), an algometer (square) and a pneumatic cuff (triangle). Both pain threshold (blue) and temporal summation (green) were assessed.

examiners on test day Two was the opposite of that on test day One (e.g. X and W), **d)** the number of test sessions were roughly evenly scheduled for the four examiners, **e)** test day Three was performed by one of the two testers chosen at random.

## Statistical analysis

**Force measurement.** Euclidean distances are presented as parametric summary statistics. Euclidean distances are analyzed using analysis of variance with participant and probe as predictor variables. The effect of the protocol (guided vs unguided) is not included, as it is eliminated by DTW.

For summary data, the number of observed pressure applications are reported as simple counts (statistical frequency) of wave peaks. Amplitude and wave period are presented as analysis of variance with participant, probe and test protocol (guided/unguided) as predictor variables.

**Sine or square wave form.** Euclidean distances and peak forces are presented as parametric summary statistics. Difference in Euclidean distance is assessed using Unpaired Wilcoxon rank sum test. Force measurement plots were inspected visually.

**Pain measurement.** To account for multiple independent variables, we analyzed pressure pain threshold data as a mixed linear regression model with pressure pain thresholds as the dependent variable and tester, day, session, repetition, anatomical site and tool as fixed effects and test subject as a random effect. Pressure pain thresholds are presented as box plots (non-parametric distribution) subdivided by independent predictor variables.

Correlation between pressure pain threshold using the probe and the algometer and cuff, respectively are presented as Pearsons correlation coefficients.

Summary statistics are presented for pain intensity with single and repeated pressure. The difference (temporal summation) is analyzed using paired Student's t-test. The correlation between temporal summation using the probe and the cuff is presented as Pearson's correlation coefficient.

## Ethics

The study was submitted for review and approved by the Regional Health Sciences Committee for the Region of Southern Denmark (S-20180045). All participants gave written informed consent prior to testing.

## Results

### Force measurement

Twenty four participants completed 189 tests in the fall of 2021 –see Table 1. The unguided protocol with no metronome sound cues was tested 101 times, and the guided protocol 88 times.

**Table 1. Eulerian distances.**

| Probe | N | Min | Max | Median | Q1 | Q3 | IQR | MAD | Mean | SD | SEM | CI |
|---|---|---|---|---|---|---|---|---|---|---|---|---|
| Black (7.4 kg) | 48 | 59.5 | 353.8 | 124.9 | 96.7 | 139.8 | 43.1 | 39.9 | 128.7 | 52 | 7.5 | 15.1 |
| Gold (5.8 kg) | 48 | 55.5 | 361.7 | 119.3 | 104.5 | 148.4 | 43.9 | 26.1 | 135.3 | 65.3 | 9.4 | 19 |
| Red (2.1 kg) | 46 | 44.7 | 573.1 | 94.2 | 60.4 | 118.9 | 58.6 | 45.9 | 107.8 | 84.2 | 12.4 | 25 |
| Silver (4.1 kg) | 47 | 47 | 333.9 | 110.5 | 82.5 | 134.2 | 51.8 | 42.4 | 117.6 | 53.6 | 7.8 | 15.8 |
| All | 189 | 44.7 | 573.1 | 112.4 | 81.9 | 137.6 | 55.7 | 40 | 122.5 | 65.2 | 4.7 | 9.4 |

Summary statistics of Eulerian distances (N = count, Min = minimum, Max = maximum, Median = median, Q1 = first quartile, Q3 = third quartile, IQR = inter-quartile range, MAD = median absolute deviation, Mean = mean, SD = stand deviation of mean, SEM = standard error of mean, CI = confidence interval of mean).

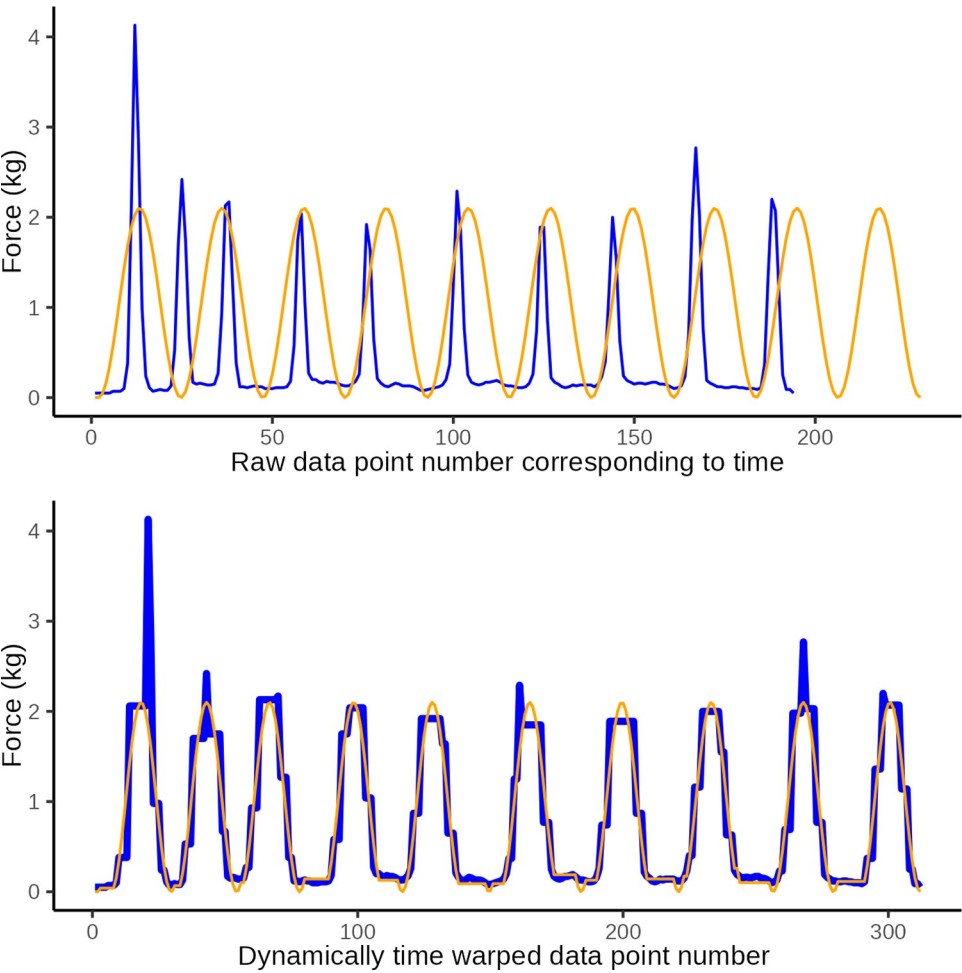

**Fig 10. Lowest Eulerian distance.** Plot of observed data (blue) with the smallest Euclidean distance (44.73) and the ideal target sine wave (orange) by an unpracticed first-time tester. The top graph illustrates the original data and the lower graph illustrates the same data after dynamic time warping (y-axis: force in kg, x-axis: observation number).

**Euclidean distance.**   Summary statistics for the Euclidean distances are presented in Table 1. Analysis of variance revealed that probe, test protocol (guided vs unguided) and their interaction were not significant predictors of Euclidean distance ($0.18 \leq P \leq 0.62$).

The tests with the lowest, median and highest observed Euclidean distance (44.73, 112.40, and 573.12, respectively) are presented in Figs 10–12.

**Summary data analyses.**   The number of pressure applications (wave peak count) are summarized in Table 2.

**Wave period.**   The distribution of duration of pressure applications cycles (wave periods) are illustrated in Fig 13 and summarized in Table 3.

**Amplitude.**   The distribution of the mean difference between peak force application and target force is illustrated in Fig 14 and summarized in Table 4.

## Sine or square wave form

**Euclidean distance.**   Summary statistics of Euclidean distances of 20 tests (of 10 force applications) by a practiced tester grouped by ideal time-force function (square or sine wave) are presented in Table 5. Euclidean distance was generally smaller for the sine wave function

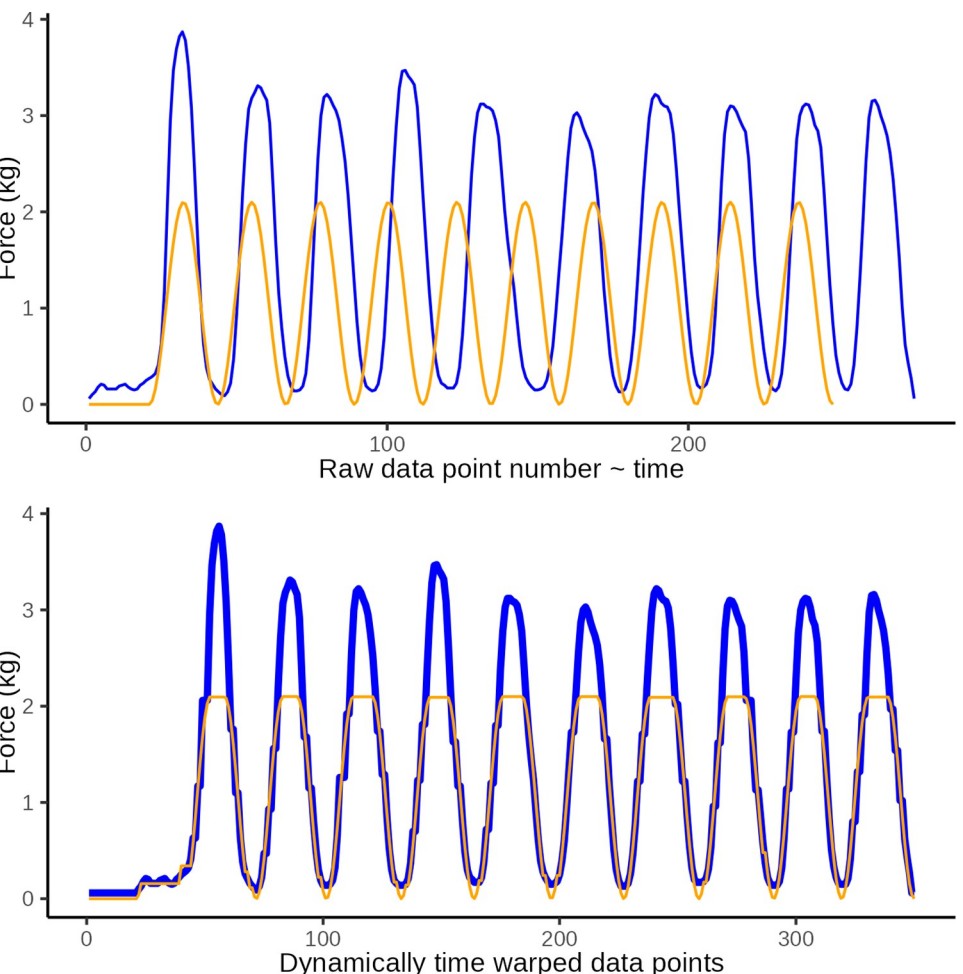

**Fig 11. Median Eulerian distance.** Plot of observed data (blue) with the median Euclidean distance (112.40) and the ideal target sine wave (orange) by an unpracticed first-time tester. The top graph illustrates the original data and the lower graph illustrates the same data after dynamic time warping (y-axis: force in kg, x-axis: observation number).

than the square wave. Unpaired Wilcoxon test confirmed this difference was highly significant ($P<0.0001$).

Visual inspection of plots of the raw data and dynamic time warped data, also suggested poorer performance with a square wave target due to inability to stay close to the square curve during changes in force.

For illustrations of the two force application series with the lowest Euclidean distance, see Figs 15 and 16.

**Peak force.** Summary statistics of peak force of 20 tests (of 10 force applications) by a practiced tester grouped by ideal time-force function (square or sine wave) are presented in Table 6 and illustrated in Fig 17. Peak force was generally closer to the target force of 4.0 Kg for the sine wave function than the square wave. Unpaired Wilcoxon test confirmed this difference was highly significant ($P<0.0001$).

## Pain measurement

Twenty seven healthy participants were recruited in 2018 between June 18th and September 30th, all were senior year university masters students.

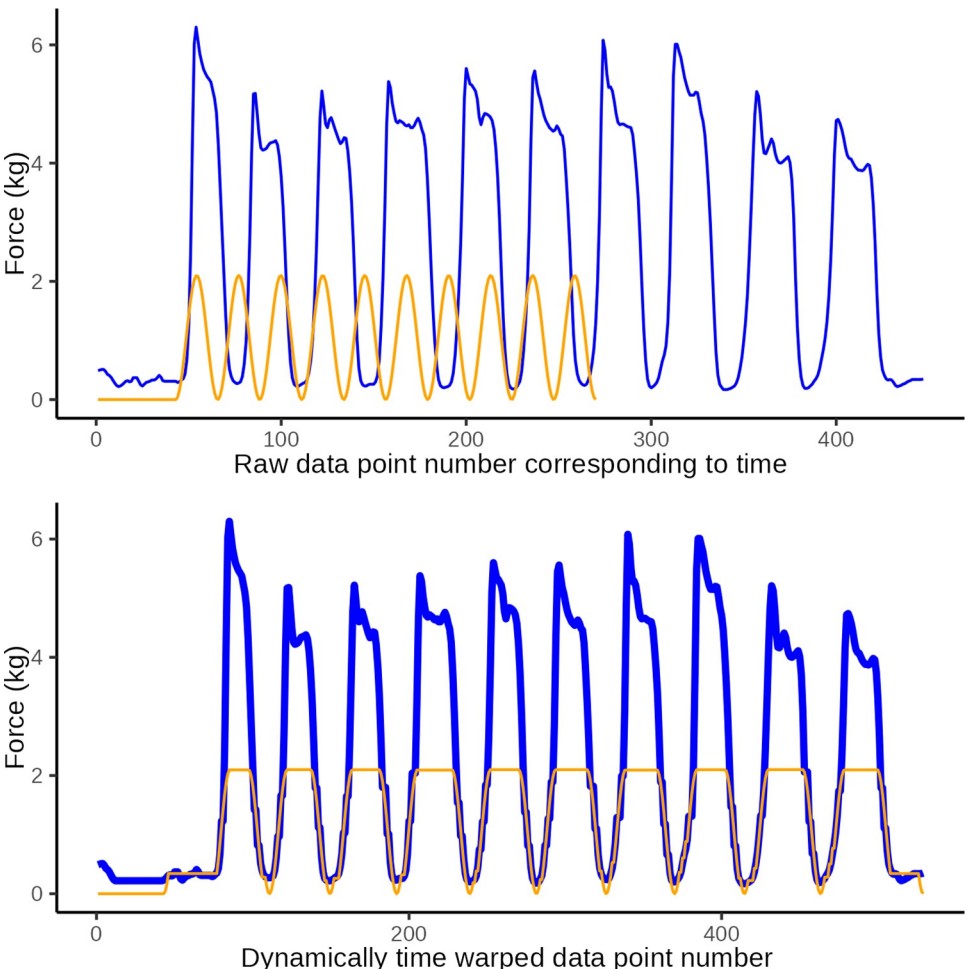

**Fig 12. Highest Eulerian distance.** Plot of observed data (blue) with the largest Euclidean distance (573.12) and the ideal target sine wave (orange) by an unpracticed first-time tester. The top graph illustrates the original data and the lower graph illustrates the same data after dynamic time warping (y-axis: force in kg, x-axis: observation number).

**Pressure pain threshold.** Boxplots of the distributions of pressure pain thresholds are presented in Fig 18.

A summary of the mixed linear regression model is presented in Supporting Information in addition to a histogram and Q-Q plot of residuals. Visual inspection did not reveal any obvious deviation from homoscedasticity or normality. Fifty-eight percent of the total variance was attributable to the true PPT variance between tested participants. An analysis of variance is presented in Table 7.

**Table 2. Number of applications.**

| protocol | 9 | 10 | 11 | 12 | 13 | 18 | 19 | 21 |
|---|---|---|---|---|---|---|---|---|
| Guided | 0 | 52 | 31 | 0 | 1 | 1 | 1 | 2 |
| Unguided | 5 | 90 | 5 | 1 | 0 | 0 | 0 | 0 |

Frequency table of number of applied pressures (target = 10) for the unguided protocol without audio cue and the guided protocol with tick/tock audio cue. Column headers list number of pressure applications and cells the number of instances observed.

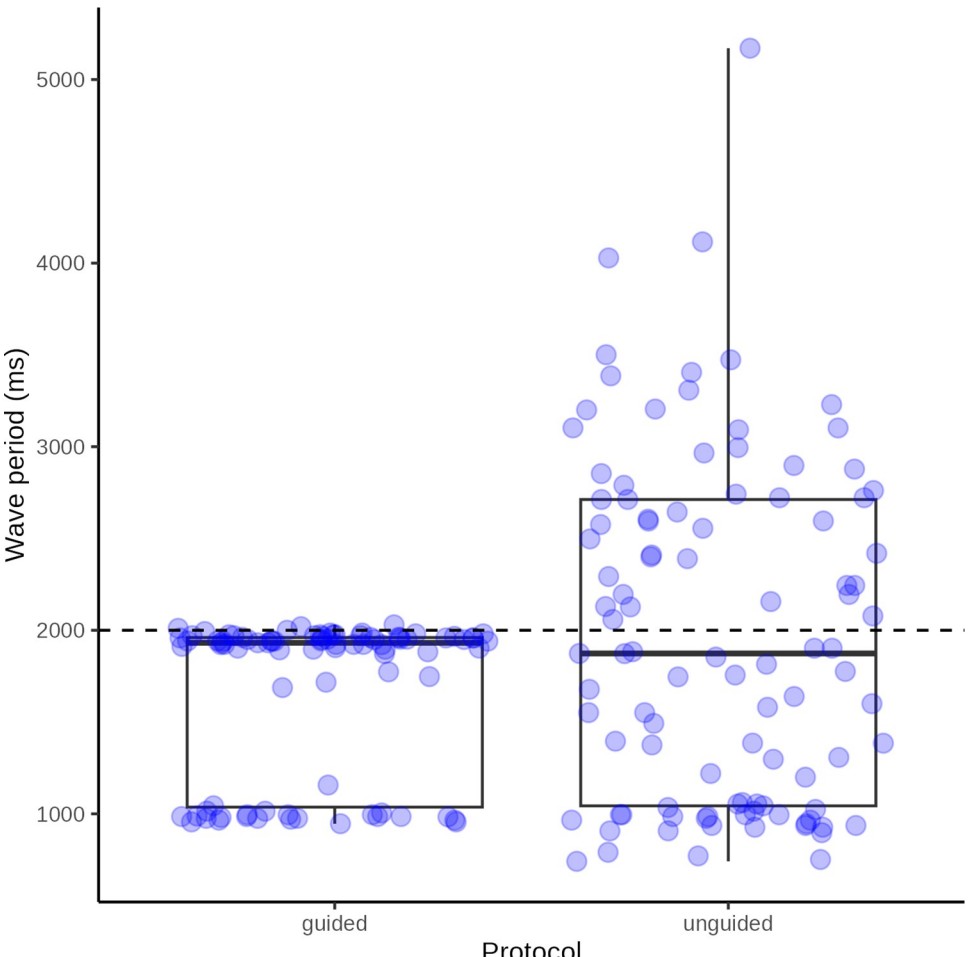

**Fig 13. Pressure application duration.** Box plot with superimposed jitter plot of raw data: mean wave period of pressure application cycles grouped by test protocol. The dashed line represent the ideal target wave period of 2.0 s.

Of the 3 categorical predictors (participant id not included), the effects sizes were found to be moderate (-0.56) for test tool, moderate (0.57) for anatomic test site and negligible (0.2) for repetition, respectively.

**Pressure pain thresholds – correlation between probe, algometer and cuff.** We found a highly significant strong correlation (Pearsons $\rho = 0.63$, $P < 0.0001$) between pressure pain thresholds measured using the hand-held algometer and the set of pressure probes.

**Table 3. Wave periods.**

| Protocol | N | Min | Max | Median | Q1 | Q3 | IQR | MAD | Mean | SD | SEM | CI |
|---|---|---|---|---|---|---|---|---|---|---|---|---|
| Guided 1 | 24 | 946.2 | 1157 | 985.3 | 974.4 | 995 | 20.6 | 14.6 | 991.7 | 41.1 | 8.4 | 17.3 |
| Guided 2 | 64 | 1688 | 2030 | 1950 | 1923 | 1971 | 47.6 | 34.7 | 1936 | 62.3 | 7.8 | 15.6 |
| Unguided | 101 | 741.3 | 5171 | 1874 | 1044 | 2712 | 1669 | 1243 | 1981 | 925.2 | 92.1 | 182.6 |

Summary statistics of observed wave period in milliseconds for the sound cue guided (divided into observations close to 1 and 2 seconds, respectively) and unguided test protocols (N = count, Min = minimum, Max = maximum, Median = median, Q1 = first quartile, Q3 = third quartile, IQR = inter-quartile range, MAD = median absolute deviation, Mean = mean, SD = stand deviation of mean, SEM = standard error of mean, CI = confidence interval of mean).

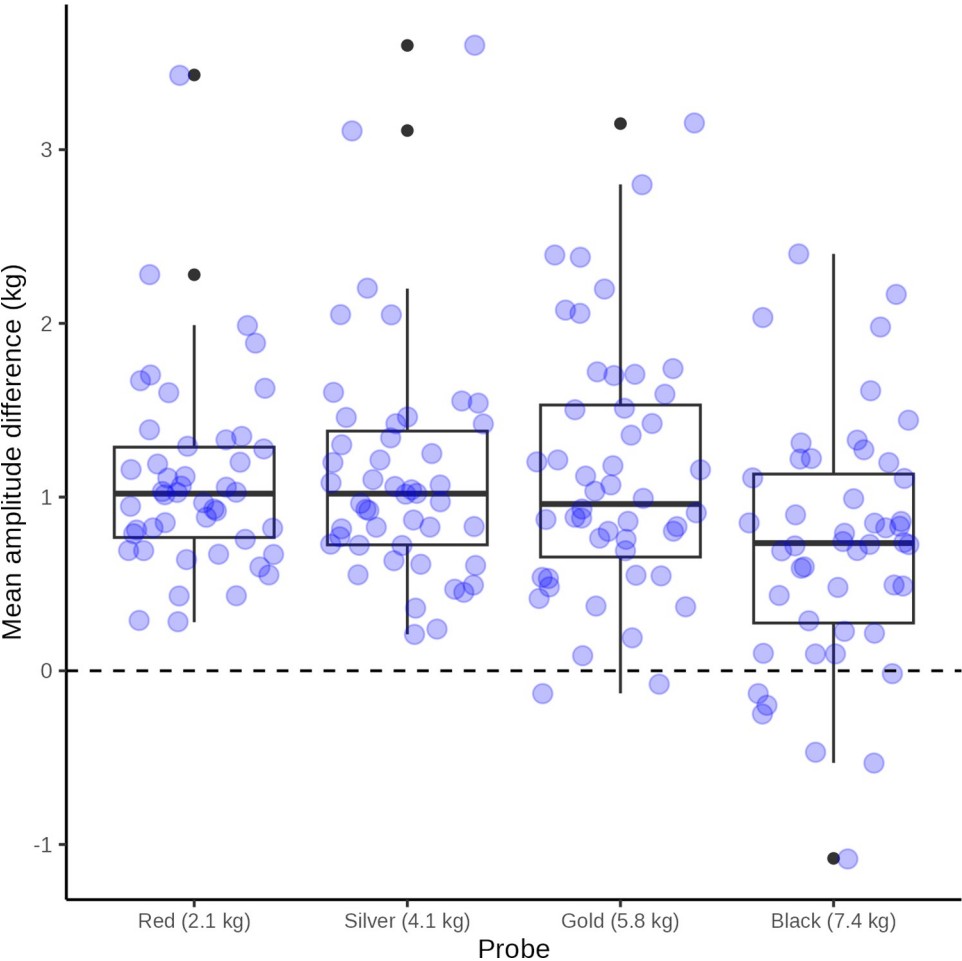

**Fig 14. Force application peak.** Box plot with superimposed jitter plot of raw data: mean difference between observed force amplitudes and target maximum force application for the four probes. For clarity, the jitter plot data points represent the mean of the 10 raw data points in each test series. The dashed line represent the ideal difference in amplitude of 0 kg.

We found a highly significant, weak correlation (Pearsons $\rho = 0.31$, $P<0.0001$) between pressure pain threshold measured using the set of pressure probes and the cuff algometer, on the lower leg.

**Table 4. Wave peaks.**

| Probe | N | Min | Max | Median | Q1 | Q3 | IQR | MAD | Mean | SD | SEM | CI |
|---|---|---|---|---|---|---|---|---|---|---|---|---|
| Black (7.4 kg) | 48 | -1.1 | 2.4 | 0.7 | 0.3 | 1.1 | 0.9 | 0.7 | 0.7 | 0.7 | 0.1 | 0.2 |
| Gold (5.8 kg) | 48 | -0.1 | 3.1 | 1 | 0.7 | 1.5 | 0.9 | 0.6 | 1.1 | 0.7 | 0.1 | 0.2 |
| Red (2.1 kg) | 46 | 0.3 | 3.4 | 1 | 0.8 | 1.3 | 0.5 | 0.4 | 1.1 | 0.6 | 0.1 | 0.2 |
| Silver (4.1 kg) | 47 | 0.2 | 3.6 | 1 | 0.7 | 1.4 | 0.7 | 0.4 | 1.1 | 0.7 | 0.1 | 0.2 |
| All | 189 | -1.1 | 3.6 | 0.9 | 0.6 | 1.3 | 0.7 | 0.5 | 1 | 0.7 | 0 | 0.1 |

Summary statistics of difference between observed wave peak force and target peak force in kg for the 4 probes (N = count, Min = minimum, Max = maximum, Median = median, Q1 = first quartile, Q3 = third quartile, IQR = inter-quartile range, MAD = median absolute deviation, Mean = mean, SD = stand deviation of mean, SEM = standard error of mean, CI = confidence interval of mean).

**Table 5. Eucledian distances.**

| Ideal | N | Min | Max | Median | Q1 | Q3 | IQR | MAD | Mean | SD | SEM | CI |
|-------|---|-----|-----|--------|----|----|-----|-----|------|----|-----|----|
| Sine | 10 | 45.1 | 81.8 | 63.5 | 52.3 | 71.9 | 19.5 | 16.2 | 63 | 13.4 | 4.2 | 9.6 |
| Square | 10 | 92.8 | 133.3 | 121.7 | 116.2 | 124.7 | 8.5 | 7.4 | 118.4 | 12.5 | 4 | 8.9 |

Summary statistics of Eucledian distance between observed force application and ideal target curve (sine wave vs square wave) with different audio cues (N = count, Min = minimum, Max = maximum, Median = median, Q1 = first quartile, Q3 = third quartile, IQR = inter-quartile range, MAD = median absolute deviation, Mean = mean, SD = stand deviation of mean, SEM = standard error of mean, CI = confidence interval of mean).

**Temporal summation of pressure pain.** Summary statistics of pressure pain intensity and temporal summation of pressure pain intensity are presented in Table 8. The difference in pain intensity between a single and repeated pressure was highly significant (paired student t-test, $P<0.0001$)

**Temporal summation of pressure pain–correlation between probe and cuff.** We found no significant correlation (Pearsons $\rho$ = -0.11, $P \geq 0.05$) between temporal summation of pressure pain threshold measured using the pressure probe on the Infra Spinatus muscle and the cuff algometer, on the lower leg.

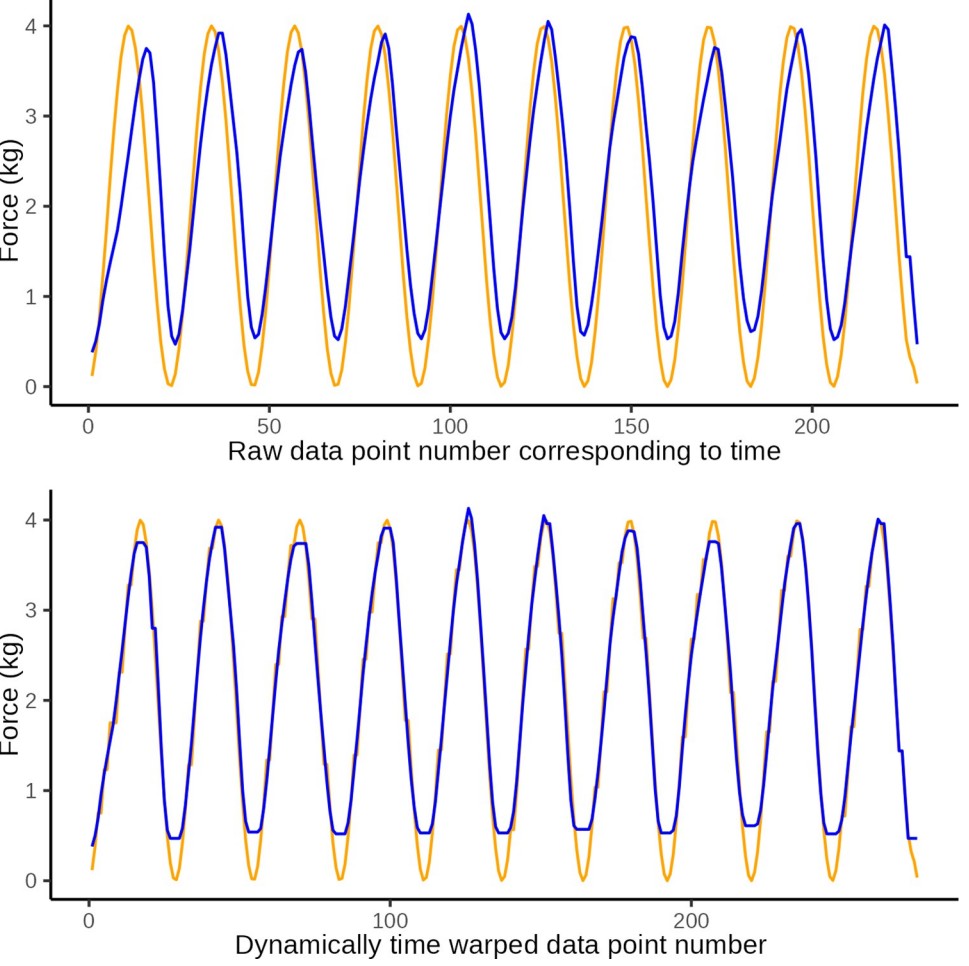

**Fig 15. Best sine wave approximation.** Plot of observed data (blue) with smallest Euclidean distance (43.47) of sine wave approximations and the ideal sine wave (orange) by a practiced tester (y-axis: force in kg, x-axis: observation number).

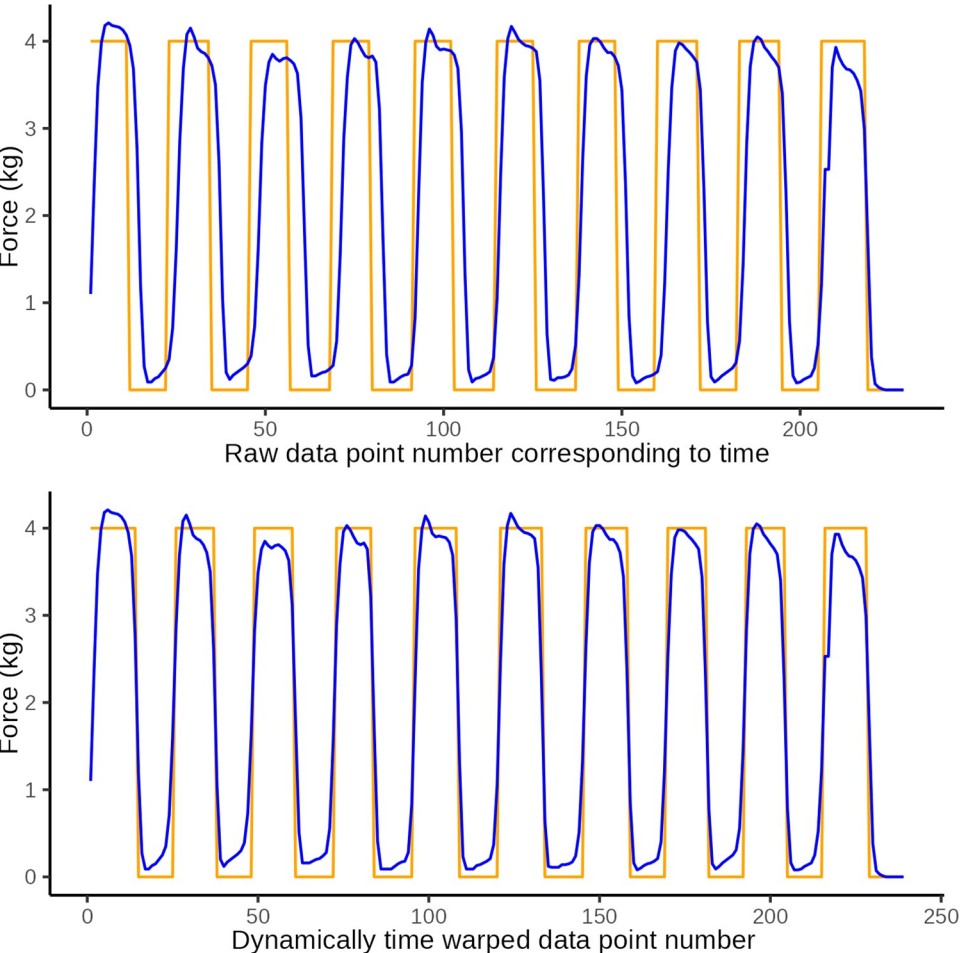

**Fig 16. Best square wave approximation.** Plot of observed data (blue) with smallest Euclidean distance (92.89) of square wave approximations and the ideal square wave (orange) by a practiced tester (y-axis: force in kg, x-axis: observation number).

## Discussion

A mechanical pressure probe was developed and assessed as a tool for quantitative sensory testing with particular emphasis on the temporal summation of deep pressure stimuli.

An auditory guide proved necessary to ensure the correct number and rate of pressure applications.

**Table 6. Peak force.**

| Ideal | N | Min | Max | Median | Q1 | Q3 | IQR | MAD | Mean | SD | SEM | CI |
|---|---|---|---|---|---|---|---|---|---|---|---|---|
| Sine | 100 | 3.4 | 5.1 | 4 | 3.9 | 4.2 | 0.3 | 0.2 | 4 | 0.2 | 0 | 0 |
| Square | 100 | 3.7 | 5 | 4.2 | 4 | 4.3 | 0.3 | 0.2 | 4.2 | 0.2 | 0 | 0 |

Summary statistics of observed peak force between observed force application and ideal target curve (sine wave vs square wave) with different audio cues (N = count, Min = minimum, Max = maximum, Median = median, Q1 = first quartile, Q3 = third quartile, IQR = inter-quartile range, MAD = median absolute deviation, Mean = mean, SD = stand deviation of mean, SEM = standard error of mean, CI = confidence interval of mean).

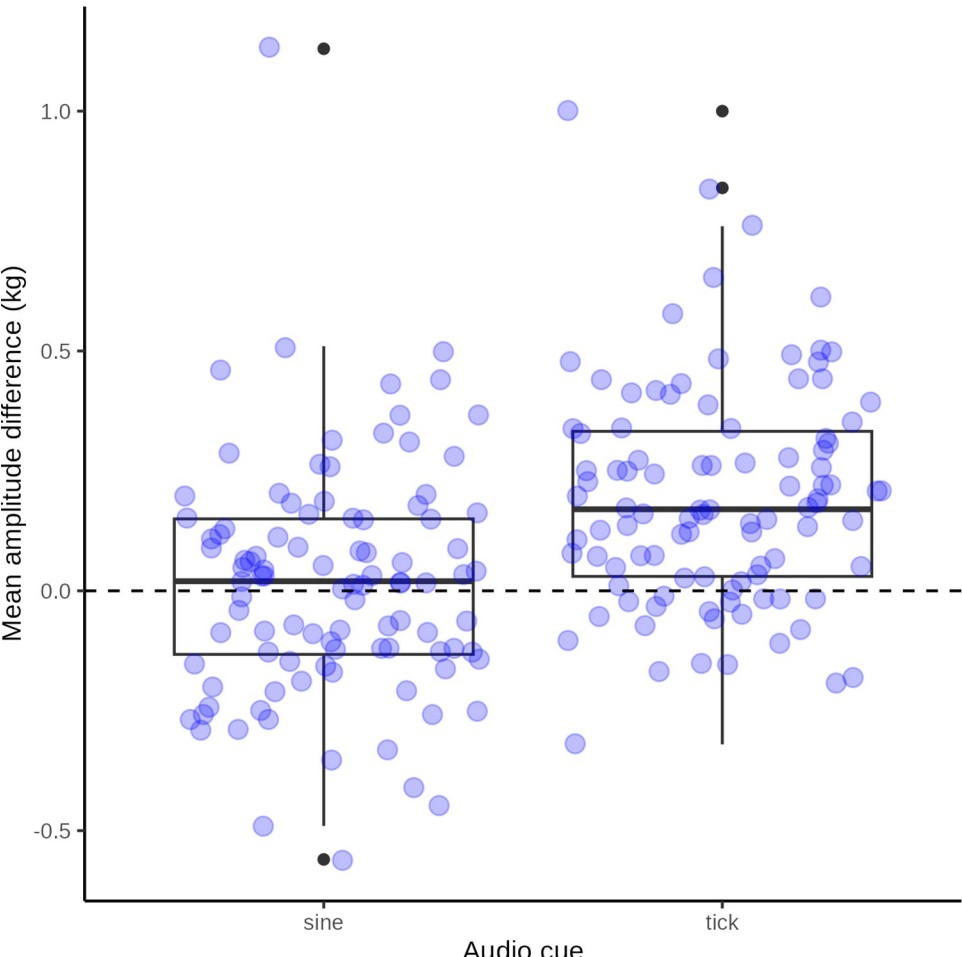

**Fig 17. Force application difference.** Box plot with superimposed jitter plot of raw data: mean difference between observed force amplitudes and the target maximum force application of 4 kg probes for a practiced tester approximating a sine wave form and a square wave form. The dashed line represent the ideal difference in amplitude of 0 kg.

In technical terms, force application aiming for a sine wave was more true to the ideal time-force curve than a square wave. Whether this has implications for the pain experience is not known.

A test platform is necessary for periodic re-calibration and for rehearsing correct use and for in-house reliability testing.

A flat circular probe head caused excessive superficial strain compared to a spherical head. We recommend a spherical probe head in line with previous recommendations [11].

We release the probe design details, audio file, testing platform design details and software into the public domain.

## Force measurement

The force measurement platform was programmed to read sensor data continuously rather than enforcing a fixed frequency of sampling, which resulted in a small degree of variation in sampling rates. This is unlikely to make any difference in the current results, but for future implementation it could be useful to upgrade the acquisition software in order to synchronize

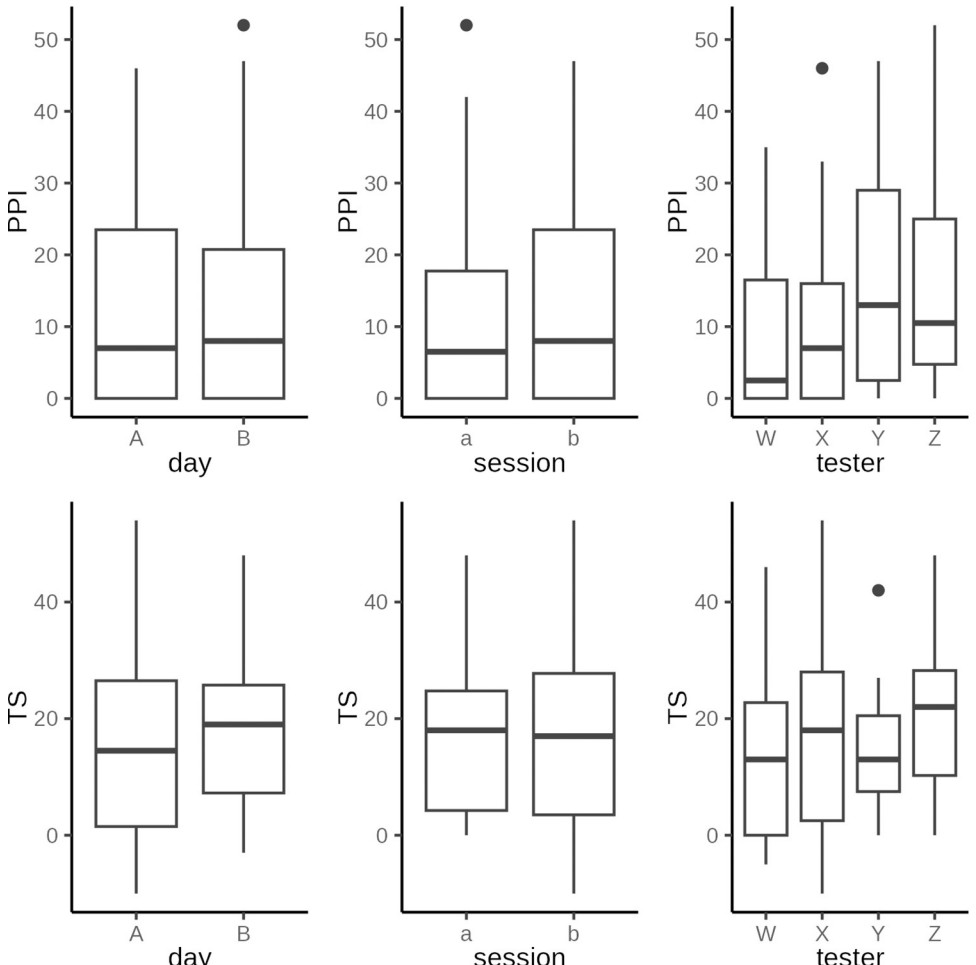

**Fig 18. Pressure pain thresholds.** Boxplot of pressure pain thresholds by independent variables: Two test days, each with two session, two repetitions per session, with different testers, with both tools and both test sites.

**Table 7. ANOVA.**

|  | Df | Sum Sq | Mean Sq | F value | Pr(>F) |
|---|---|---|---|---|---|
| id | 26 | 2341 | 90 | 48.3 | 0 |
| day | 1 | 1.4 | 1.4 | 0.8 | 0.4 |
| session | 1 | 3 | 3 | 1.6 | 0.2 |
| tester | 3 | 9.9 | 3.3 | 1.8 | 0.2 |
| tool | 1 | 340.1 | 340.1 | 182.6 | 0 |
| test_site | 1 | 354 | 354 | 190 | 0 |
| rep_number | 1 | 46 | 46 | 24.7 | 0 |
| Residuals | 829 | 1544 | 1.9 | NA | NA |

Analysis of variance of pressure pain thresholds. The tested participant (id), the test tool (algometer or probe), the anatomical test site (lower back or lower leg) and the repetition number (1 or 2) were found to be significant predictors of pressure pain threshold.

**Table 8. Pressure pain intensity.**

| Variable | N | Min | Max | Median | Q1 | Q3 | IQR | MAD | Mean | SD | SEM | CI |
|---|---|---|---|---|---|---|---|---|---|---|---|---|
| Single pressure | 135 | 0 | 52 | 8 | 0 | 22 | 22 | 11.9 | 12.7 | 13.3 | 1.1 | 2.3 |
| Repeated pressure | 135 | 0 | 85 | 29 | 13 | 44.5 | 31.5 | 23.7 | 29.3 | 21.3 | 1.8 | 3.6 |
| Temp. sum. | 135 | -10 | 54 | 17 | 5 | 25 | 20 | 14.8 | 16.7 | 13.9 | 1.2 | 2.4 |

Summary statistics of pressure pain intensity (0–100 VAS) with one pressure application, after ten repeated pressure applications and their difference (temporal summation) (N = count, Min = minimum, Max = maximum, Median = median, Q1 = first quartile, Q3 = third quartile, IQR = inter-quartile range, MAD = median absolute deviation, Mean = mean, SD = stand deviation of mean, SEM = standard error of mean, CI = confidence interval of mean).

sampling with an accurate time base generated by a hardware timer (present inside the ATmega328 processor).

The rationale behind the setup of the force measurement tests was not to achieve the most convincing results. Rather it was to learn which aspects of the use of pressure probes were likely to present problems and how to mitigate them. In this regard, we were successful in identifying several caveats:

**Adherence to test protocol.** Despite clear instructions provided by pre-recorded video, a small number of participants misunderstood the testing protocols fundamentally and applied either twice the required number of pressure applications at twice the rate, or conversely half. This could have been discovered a-priori simply by implementing a supervised practice session.

We recommend that instructions in the use of pressure probes, be supplemented by practice sessions supervised by an experienced tester.

**Rate of pressure application.** Even with a minimum of rehearsal, no feedback and no audio cues, volunteer testers were able to strike a rate of pressure applications close to the target wave period of 2 s on average. However, closer examination revealed large variations ranging from less than 1 to more than 5 s. Conversely, when testers were guided by an audio cue, the large majority were well able to maintain the correct rate with very little variation. The small number of testers who, despite an audio cue, had a rate of application twice that of the target rate, illustrates the need for sufficient instruction of testers.

We recommend the use of a sound cue to ensure that testers can reliably maintain the correct rate of pressure application with little variation and we recommend that instructions in the use of pressure probes, be supplemented by practice sessions supervised by an experienced tester.

**Number of pressure applications.** It was surprising, that more testers performed the correct number of pressure applications per series without a sound cue, than with a cue. The most common incorrect number of pressure applications with a sound cue was 11, i.e. one pressure application too many. This suggests to us, that without a sound cue, tester are forced to count the number of pressure applications. Conversely, when guided by a sound cue testers simply 'go along' without counting and once in the rhythm of it, may perform an extra pressure application and only realize this too late.

We recommend the use of a sound cue that takes this into account by clearly signaling when to stop pressure applications.

## Approximating a sine or square wave ideal function

From the post-hoc tests of sine wave vs square wave approximations, it seems that human testers are better at approximating a sine wave.

This conclusion is based on only a single tester and a limited data set, but the finding is hardly surprising: Approximating a square wave function requires the tester to repeatedly make very sudden changes in the applied force and then to maintain that pressure at a constant level. This alternation between instantaneous changes in force followed by sustained and constant force application is impossible to perfect and difficult to approximate. Conversely, with only ten practice sessions, it is much easier to exert the constantly fluctuating force which approximates a sine wave.

Whether the better fidelity of the sine wave approximation makes any difference in pain experience for a test person is unclear and future studies should examine this.

The tendency to overshoot the target force by ~1 kg may be a consequence of attempting to mimic the square wave form with its instantaneous increases in force. In any case, with practice in the post hoc test the difference in overshoot between sine and square wave approximations was very small, albeit significant.

We recommend using a sine wave ideal force function.

## Pain measurements

In line with previous publications on pressure pain, the test-retest and inter-tester reliability were good [14–17]: Tester, test day and test session did not significantly influence pressure pain thresholds and the effect of test repetition (1st vs. 2nd test in short succession) was negligible.

It was not surprising, that the pressure pain thresholds were found to be significantly and moderately lower over the superficial Tibialis Anterior muscle, compared to the deep lumbar paraspinal muscles, which are covered by the thick thoracolumbar fascia. Generally speaking, pressure pain thresholds can be expected to differ between test sites [18].

Pressure pain thresholds were higher when using a probe compared to the algometer. As the probe and algometer heads were identical we suggest the differences in test procedures can explain this observation: using the split-middles method with mechanical pressure probes will by design identify the lowest supra-threshold pressure rather than the threshold itself. This will be quantified with an accuracy determined by the span in pressure between probes, which in the current study was 1 kg. In other words, determining a pressure pain threshold using pressure probes will tend to overestimate it by as much as the span between probes and this is exactly what we observed in the current data. Also, the split-middles method necessitates repeated pressure applications to narrow down the threshold and such repeated pressure applications may influence the pain perception by temporal summation. By contrast, using an algometer requires only a single gradually increasing pressure application.

We recommend continued use of an algometer to assess pressure pain thresholds rather than a split middle methods approach using mechanical probes. We also note however, that the strong correlation between them speaks to the validity of mechanical probes as tools for evoking pressure pain.

In a study of unilaterally induced *delayed onset muscle soreness* (DOMS) of the Trapezius muscles, Nie et al. [7] reported temporal summation of pressure pain when using a computer controlled algometer in a test paradigm not dissimilar to the current pressure probe setup. The authors observed that temporal summation depended on the frequency of repeated pressure stimuli. In a more recent publication using the same probes as described in this paper, we found significant increases in pressure pain in healthy individuals following delayed onset muscle soreness (DOMS), and found that temporal summation of pressure pain was not affected by DOMS [19]. In a previous publication we found that temporal summation of pain

with mechanical probes, may be a better choice for test stimuli in a conditioned pain modulation paradigm [20] compared to pressure pain thresholds.

The lack of a significant correlation in the current data between temporal summation of pain using the probes and the cuff on the lower leg is note-worthy. We suggest this relates to important differences between the two test methods. Specifically, that two different muscles were tested (upper limb vs lower limb) and that the mechanical pressure applied was applied differently. The probe force is focal and direct whereas the cuff is more diffuse and circumferential. A lack of coherence between different test setups for conditioned pain modulation has previously been reported and perhaps different tests simply represent different pain perceptions [21]. As stated it is unclear how this affects the validity of the QST results and future studies should examine this.

## Public domain

We have released the design specification along with relevant software and instructions into the public domain. This includes the ability to produce, modify and distribute the pressure probes freely. We urge anyone who does modify and distribute probes based on our design to do so in the same spirit and release the specifications into the public domain.

It is our hope that this may help facilitate clinically relevant pain research in different contexts, including low-income countries.

Please refer to the following web page for further details: http://www.smerteforskning.dk/qst/.

## Conclusion

Using the design specifications we have released into the public domain, it is possible to manufacture mechanical pressure probes for experimental pain research at very low cost.

The probes can and should be re-calibrated at regular intervals using a pressure test platform and are then suitable for assessing pain intensity with sustained or repeated mechanical pressure, but not for determining threshold values.

The validity of the probes as a tool for temporal summation of pressure pain is improved by regular re-calibration, by using an audible cue approximating a sine wave and by ensuring sufficient prior practice using the equipment.

## Supporting information

**S1 File. Summary of the mixed linear regression model including a histogram and Q-Q plot of residuals.**
(DOCX)

**S1 Data.**
(ZIP)

## Acknowledgments

We would like to acknowledge Henrik Baare Olsen and Tue Skallgaard from the Institute of Sport Science and Clinical Biomechanics, University of Southern Denmark, for their assistance with design and manufacturing of the calibration platform.

## Author Contributions

**Conceptualization:** Søren O'Neill.

**Data curation:** Søren O'Neill, Casper Glissmann Nim, Natalie Hong Siu Chang.

**Formal analysis:** Søren O'Neill.

**Investigation:** Søren O'Neill.

**Methodology:** Søren O'Neill.

**Software:** Søren O'Neill.

**Writing – original draft:** Søren O'Neill.

**Writing – review & editing:** Søren O'Neill, Casper Glissmann Nim, Natalie Hong Siu Chang.

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
