## [Decision Letter · Decision Letter 0]

16 Aug 2023

PONE-D-23-13550Validation of a spring loaded probe for single and repeat pressure pain testing, including public domain specifications for design and manufacturePLOS ONE

Dear Dr. Søren O’Neill,

Thank you for submitting your manuscript to PLOS ONE. After careful consideration, we feel that it has merit but does not fully meet PLOS ONE’s publication criteria as it currently stands. Therefore, we invite you to submit a revised version of the manuscript that addresses the points raised during the review process.

ACADEMIC EDITOR: 

Please take into account the reviewers' recommendations regarding the following issues:

Include details about the relationship or contrast of temporal summation pain (TSP) and pressure pain threshold (PPT) based on literature studies.Add a comment about the possibility that the 3D-printed plastic tube (a') influences the applied force?Check if it is appropriate to consider the subjects who did not understand the protocol as outliers and exclude them from the statistics in Table 3.Justify the choice of the unpaired variant for the Wilcoxon test.Add a comment about how to use the sum of Euclidean distances to compare the evolution of pressures in the case of recordings with different lengths and amplitudes.Specify the exact meaning of the expression "repeated pressure" from Table 8.Review the use of the terms "wavelength" and "pressure"Please solve the technical problems related to Figure 9 which is the same as Figure 8, and many figures which are shown incompletely (6,7,10,12...)

We look forward to receiving your revised manuscript.

Kind regards,

Livia Petrescu

Academic Editor

PLOS ONE

Comments from Journal Office: We note that one or more reviewers has recommended that you cite specific previously published works. As always, we recommend that you please review and evaluate the requested works to determine whether they are relevant and should be cited. It is not a requirement to cite these works. We appreciate your attention to this request.

Journal Requirements:

We the authors declare no conflicts of interest in the preparation of this manuscript.

The tool described in the manuscript may have some commercial potential, but as

stated in the manuscript, we release the design specifications into the public domain and do so with the intention of making it easily and widely available for the pain research community.

This includes the potential for anyone to produce and sell the tool for profit without any obligations (royalties or otherwise) to us. We the authors have no plans to sell the apparatus for profit.

Reviewers' comments:

Reviewer's Responses to Questions

**Comments to the Author**

1. Is the manuscript technically sound, and do the data support the conclusions?

Reviewer #1: Yes

Reviewer #2: Partly

2. Has the statistical analysis been performed appropriately and rigorously? 

Reviewer #1: Yes

Reviewer #2: Yes

3. Have the authors made all data underlying the findings in their manuscript fully available?

Reviewer #1: Yes

Reviewer #2: Yes

4. Is the manuscript presented in an intelligible fashion and written in standard English?

Reviewer #1: Yes

Reviewer #2: Yes

5. Review Comments to the Author

Reviewer #1: An interesting study on validation of a simple mechanical test apparatus to assess the temporal summation of deep pressure pain. Therefore, temporal summation and pressure pain are two key measuring issues of this study. The authors initiated as “temporal summation of pressure pain is technically more challenging than simple pressure pain thresholds”. However, simple relationship or contrast of temporal summation pain (TSP) and pressure pain threshold (PPT) are not established from scientific literature in this manuscript. The following articles may assist the authors to contrast TSP and PPT in clinical scenario:

1. A study (https://pubmed.ncbi.nlm.nih.gov/33316140/) indicated a clinical presentation with psychological factors (one dimension of risk) is not associated with high-PPT/sensitivity (another dimension of risk).

2. The authors can also improve discussion by taking evidence from this paper ( https://pubmed.ncbi.nlm.nih.gov/30648781/) by indicating the differential predictive value of dynamic (TSP) versus static (PPT) measures of pain sensitivity as well as the unique value of evoked measures (and the potential limitations of psychological factors) in predicting functional outcome.

3. A commentary article, available at https://journals.lww.com/clinicalpain/Citation/2021/04000/Movement_evoked_Pain__MEP_.8.aspx , may help authors making a nice future direction on usefulness of QST in MSK pain mechanism including temporal summation of pressure pain.

I hope after a revision reader will be more clear on science link clinical application of TSP and PPT as well as this new evolving technique of this manuscript (temporal summation of pressure pain).

Reviewer #2: The subject of this article is interesting namelly develpoing a simple and low cost spring loaded probe for pressure paing testing as an alternative to more expensive devices.

After reading and analyzing the paper, I can make the following observations:

- Can the 3D printed plastic tube (a') influence the applied force? If the spring is compressed enough for this tube to touch the handle of the probe, the spring has no further role to control of the applied force (force is transmitted directly from handle to the probe head).

- Sampling time of the Arduino based platform can be made very precise by proper design of the acquisition software (using a hardware timer and placing acquisition sequence inside the timer interrupt handling routine). However data presented in the article indicates a small non-uniformity of sampling times (2.46 ms relative to 87.78 ms) which probably has no significant effect on measurment results. Acquisition software redesign can be mentioned as future improvements of the test platform.

- In "Hardware and software for testing" section, there is a quite large part that describes how DTW works (includes Figures 6 to 8). In my opinion this part is not required (information about DTW can be given by a reference) and at first read it can create confusion especially by using terms "constructed examples/simulated series of observational data". Only important information as defining Euclidian distance can be kept.

- About Euclidian distance: the metrics used in eq. 1 and 2 are not simple Euclidian distance between two samples but the sum of all distances. This is equivalent to the value of the area between the two curves which is an indication of how close are these curves.

- Another observation related to the use of this sum of distances for comparison of differences between signals in various cases: The sum depends not only on the difference between signal samples but also on number of samples and values of samples. For example: two cases - one pair of signals having ED 100 and 100 samples and another pair of signals with same ED of 100 but 200 samples means that second pair of signals are more close each to other even if they have same ED. Also same ED for a pair of signals with amplitude of 10 and another pair of signals with amplitude of 20 indicated that the second pair is more close. Maybe a normalized sum of ED is better for comparison (divide sum of Euclidian distances by number of samples and amplitude of the signals).

- In case of guided tests results presented in fig. 13, thereare two clusters of wavelengths (most values are grouped around 2sec and some of them are grouped around 1 sec). I suppose that 1 sec values correspond to the testers that did not undertand the protocol. Wouldn't it be better if these values were considered as outliers and excluded from further statistics?

- Is it ok to use unpaired Wilcoxon test for sine/square comparison? Data are obtained from same tester using same tool - maibe paired version would be more apropriate?

- In table 8, Repeated pressure is reffering to the average value of the 10 repeated aplications?

- As authors indicated in the last section of "Discussion" the corelation between temporal sumation obtained by probe and cuff is unrelevant because different tools are used on different sites and it is expected to be uncorrelated. Should this correlation test be kept in the article or it can be removed?

- Be careful when using different terms:

* wavelength is a characteristic of a wave that travels through a medium. A more apropriate term is period.

* pressure in figure 5 must be replaced with force.

- Figure 9 is same as Figure 8.

- Many figures are shown incompletely (6,7,10,12...)

In conclusion, I recommend to reconsider the article after a major revision.

6. PLOS authors have the option to publish the peer review history of their article (what does this mean?). If published, this will include your full peer review and any attached files.

Reviewer #1: **Yes: **Zakir Uddin

Reviewer #2: No

---

## [Author Response · Author response to Decision Letter 0]

28 Aug 2023

---

title: "Response to Reviewers 1"

author: "S. O'Neill"

format: pdf

editor: source

---

Dear editor and reviewers

Thank you very much for a thorough review of our manuscript. We appreciate the lucid criticism and valid points raised which have aided in improving our submission.

# Review comments to the author

## Reviewer #1

We thank the reviewer for the kind words. We have read the recommended publications with interest and agree they illustrate the potential relevance of evoked pain measures. We have amended the first sentence of the introduction and included a citation of one of the papers:

> The literature on _temporal summation_ (TS) of pressure pain and its relation to clinical status is not entirely clear**, but there is evidence e.g. that experimental pain sensitivity including temporal summation of pain is predictive of clinical aspects such as fear-related activity avoidance[REF]**.

Text highlighted in bold, has been added.

The main issue relating to temporal summation in this context is that most studies employ superficial stimulation, in contrast to deep stimulation which is the subject of the current study. We already briefly mention that there is some evidence relating to deep pressure pain in chronic LBP and osteoarthrosis of the knee joint.

## Reviewer #2

**"Can the 3D printed plastic tube (a') influence the applied force? If the spring is compressed enough for this tube to touch the handle of the probe, the spring has no further role to control of the applied force (force is transmitted directly from handle to the probe head)."**

The reviewer is quite right, that once the spring has been compressed to the point where the plastic tube (a') contacts the probe handle, if the pressure is increased further, the spring will play no further role in controlling applied force. The point of including the plastic tube, is to provide the operator with tactile feedback indicating, that target force has been reached (and no further pressure should be added).

We have amended the last part of paragraph five ("In earlier experimentation with handmade prototypes,..") in Materials and methods, thus:

> In the end, using a 3D-printed hard plastic tubing of the right diameter proved both simple and efficient. It allowed for easy re-adjustment and importantly provided a tactile rather than visual clue to the tester that the correct spring compression had been reached **and no further pressure should be added, as that would overshoot the target force**. 

Text highlighted in bold, has been added.

**"Sampling time of the Arduino based platform can be made very precise by proper design of the acquisition software (using a hardware timer and placing acquisition sequence inside the timer interrupt handling routine). However data presented in the article indicates a small non-uniformity of sampling times (2.46 ms relative to 87.78 ms) which probably has no significant effect on measurment results. Acquisition software redesign can be mentioned as future improvements of the test platform."**

Thank you the suggestion which demonstrates greater insight into the technicalities of using an Arduino than we posses. We will be sure to consider this in any future projects. We agree it probably makes no difference for the data in this case and in any case, the data sampled was compared to an ideal sine wave using Eulerian which accommodates differences in sampling rate and number between curves. 

We have added the following paragraph to the Disucssion: 

> The force measurement platform was programmed to read sensor data continuously rather than enforcing a fixed frequency of sampling, which resulted in a small degree of variation in sampling rates. This is unlikely to make any difference in the current results, but for future implementation it could be useful to employ a hardware time to to force the interrupt that runs the sensor reading code.

**"In 'Hardware and software for testing' section, there is a quite large part that describes how DTW works (includes Figures 6 to 8). In my opinion this part is not required (information about DTW can be given by a reference) and at first read it can create confusion especially by using terms 'constructed examples/simulated series of observational data'. Only important information as defining Euclidian distance can be kept."**

We agree this is probably more detailed than it strictly speaking needs be, however we included it with a particular consideration in mind: The probe, as a tool for experimental pain tests, is likely to be of interest to clinical researcher studying changes in pain sensitivity in clinical conditions. We suspect that many such clinical researchers will find the analyses difficult to understand unless the principles of DTW and ED are explained in some detail. For some readers, like the reviewer, that has a deeper understanding of such analyses, it will seem superfluous, we know. We would like to keep that text as an aid for readers that do not have such understanding.

**"About Euclidian distance: the metrics used in eq. 1 and 2 are not simple Euclidian distance between two samples but the sum of all distances. This is equivalent to the value of the area between the two curves which is an indication of how close are these curves."**

Yes, eq. 1 is the root sum of squared differences (Euclidean metric). Eq. 2 is the summarized ED of these squared differences that gives the optimal warping path between the two samples. We have made some changes to this section to enhance its clarity:

> The signals were aligned using DTW, and the minimum Euclidean distances were calculated by finding the root sum of squared differences (see @eq-dtw):

> $$d_{mn}(X,Y)=\\sqrt{\\sum_{k=1}^{K} (x_{k,m} - y_{k,n})*(x_{k,m} - y_{k,n})}$$ {#eq-dtw}

> Equation @eq-dtw-optimal summarizes the Euclidean distances that provide the optimal warping path.

> $$d=\\sum_{\\substack{m \\in ix \\\\ n\\in iy}} d_{mn}(X,Y)$$ {#eq-dtw-optimal}

> For further details see MATLAB at https://se.mathworks.com/help/signal/ref/dtw.html. 

> The sum of Euclidean distances are thus an expression of the overall disparity between the observed time data and the ideal sine wave function. In other words, how similar are the curves after DTW has minimized the effects of time-disparity.

**"Another observation related to the use of this sum of distances for comparison of differences between signals in various cases: The sum depends not only on the difference between signal samples but also on number of samples and values of samples. For example: two cases - one pair of signals having ED 100 and 100 samples and another pair of signals with same ED of 100 but 200 samples means that second pair of signals are more close each to other even if they have same ED. Also same ED for a pair of signals with amplitude of 10 and another pair of signals with amplitude of 20 indicated that the second pair is more close. Maybe a normalized sum of ED is better for comparison (divide sum of Euclidian distances by number of samples and amplitude of the signals)"**

It is correct that both the number of samples and values of samples have an influence on the summarized ED. However, with a relatively consistent sample rate, we would say that this is controlled for, as we know exactly how many samples the ideal time series should have. The participants are instructed to perform the test within 20 sec. (which is also the length of the ideal curve). The case where the number of samples deviates from the ideal time series would be if the participant does not finish within 20 sec, meaning they did not follow the ideal time series. However, the normalization of this would not contribute much to the result as we use DTW to align the observed time series and the ideal time series. The discrepancy in amplitude is what we are looking for to evaluate the performance of the test. 

**"In case of guided tests results presented in fig. 13, thereare two clusters of wavelengths (most values are grouped around 2sec and some of them are grouped around 1 sec). I suppose that 1 sec values correspond to the testers that did not undertand the protocol. Wouldn't it be better if these values were considered as outliers and excluded from further statistics?"**

The reviewer is absolutely correct about the nature of the two clusters around 2sec and 1sec. We included all observations precisely to highlight the point, that using a guide can effectively help operators 'hit' the right cadence of 2sec, but as the data bears out: the operator needs sufficient instruction to actually understand the task at hand.

We have thus left the figure (Figure 13) unchanged as it illustrates two two groups nicely, but have amended the table (Table 3) to report the guided group split into those close to 1 and 2 seconds, respectively.

**"Is it ok to use unpaired Wilcoxon test for sine/square comparison? Data are obtained from same tester using same tool - maibe paired version would be more apropriate?"**

Using a paired rather than a paired wilcoxon test would not affect the conclusion. In any case, the data points are indeed all from the same tester and tool, but the data are not pair-wise inter-dependent. 

**"In table 8, Repeated pressure is reffering to the average value of the 10 repeated aplications?"**

We can appreciate how this detail was lost along the way. In fact, the VAS with repeated pressure reffers to pain after the 10th application: (Method section) "Thereafter, 10 repeated pressure applications of approximately 1 s with 1 s rest intervals were applied in the same location. After the 10th pressure application, participants indicated pain intensity using the same VAS."

We have amended the table caption:

> Summary statistics of pressure pain intensity (0-100 VAS) with one pressure application, **after** ten repeated pressure applications and their difference

Text highlighted in bold, has been added.

**"As authors indicated in the last section of "Discussion" the corelation between temporal sumation obtained by probe and cuff is unrelevant because different tools are used on different sites and it is expected to be uncorrelated. Should this correlation test be kept in the article or it can be removed?"**

We see the reviewers point. We suggest to retain this short finding, if only for completeness. Had there in fact been a noteworthe correlation between the two test procedures, we would have had to discuss it -- we therefor feel it is proper to mention that no such correlation was observed.

**"Be careful when using different terms:"**

Wavelength: Thanks you for putting us right about this - we have changed the terminology from _wavelength_ to _wave period_ throughout

Figure 5: We are a little embarrassed by this oversight and have changed figure caption and axis title as appropriate.

**"Figure 9 is the same as Figure 8"**

There seems to have been some issues with the conversion of uploaded files in the Editorial Manager platform -- we have ensured, that they have been addressed for the re-submission. Thank you for drawing our attention to the issue.

**"Many figures are shown incompletely (6,7,10,12...)"**

There seems to have been some issues with the conversion of uploaded files in the Editorial Manager platform -- we have ensured, that they have been addressed for the re-submission. Thank you for drawing our attention to the issue.

On behalf of the authors

S. O'Neill -- Aug 28th 2023

---

## [Decision Letter · Decision Letter 1]

18 Sep 2023

PONE-D-23-13550R1Validation of a spring loaded probe for single and repeat pressure pain testing, including public domain specifications for design and manufacturePLOS ONE

Dear Dr. O'Neill,

Thank you for submitting your manuscript to PLOS ONE. After careful consideration, we feel that it has merit but does not fully meet PLOS ONE’s publication criteria as it currently stands. Therefore, we invite you to submit a revised version of the manuscript that addresses the points raised during the review process.

ACADEMIC EDITOR:Please revise the article, taking into account the recommendations of reviewer 2.

We look forward to receiving your revised manuscript.

Kind regards,

Livia Petrescu

Academic Editor

PLOS ONE

Journal Requirements:

Reviewers' comments:

Reviewer's Responses to Questions

**Comments to the Author**

1. If the authors have adequately addressed your comments raised in a previous round of review and you feel that this manuscript is now acceptable for publication, you may indicate that here to bypass the “Comments to the Author” section, enter your conflict of interest statement in the “Confidential to Editor” section, and submit your "Accept" recommendation.

Reviewer #1: All comments have been addressed

Reviewer #2: (No Response)

2. Is the manuscript technically sound, and do the data support the conclusions?

Reviewer #1: Yes

Reviewer #2: Yes

3. Has the statistical analysis been performed appropriately and rigorously? 

Reviewer #1: Yes

Reviewer #2: Yes

4. Have the authors made all data underlying the findings in their manuscript fully available?

Reviewer #1: Yes

Reviewer #2: Yes

5. Is the manuscript presented in an intelligible fashion and written in standard English?

Reviewer #1: Yes

Reviewer #2: Yes

6. Review Comments to the Author

Reviewer #1: I think they addressed what I raised before for improving the manuscript.

It looks better for the readers.

Reviewer #2: After reading and analyzing the revised version of the article, I noticed that most of the previously formulated observations were addressed.

However, there are still some minor changes to make:

- To increase the accuracy of the expression, the paragraph:

"The force measurement platform was programmed to read sensor data continuously rather

than enforcing a fixed frequency of sampling, which resulted in a small degree of variation

in sampling rates. This is unlikely to make any difference in the current results, but for

future implementation it could be useful to employ a hardware time to to force the

interrupt that runs the sensor reading code."

can be modified as:

"The force measurement platform was programmed to read sensor data continuously rather

than enforcing a fixed frequency of sampling, which resulted in a small degree of variation

in sampling rates. This is unlikely to make any difference in the current results, but for

future implementation it could be useful to upgrade the acquisition software in order to

synchronize sampling with an accurate time base generated by a hardware timer (present inside

the ATmega328 processor)."

- Before equation 2 it could be useful to change the

"Equation 2 summarizes the Euclidean distances that provide the optimal warping path"

phrase with:

"Equation 2 describes the way to obtain the optimal warping path by minimization of the the Euclidean distance"

- In Discussion section at "Rate of pressure application" paragraph it could be useful to add a comment about

the results presented in Figure 13 and Table 3 indicating that the operator needs sufficient instruction to actually understand the task at hand.

In conclusion, I recommend to accept the article after minor revisions according to previous observations.

7. PLOS authors have the option to publish the peer review history of their article (what does this mean?). If published, this will include your full peer review and any attached files.

Reviewer #1: **Yes: **Dr. Zakir Uddin, PhD (McMaster), PDF (McGill)

Reviewer #2: No

---

## [Author Response · Author response to Decision Letter 1]

20 Sep 2023

Please find response to reviewers attached as a pdf file and inserted below as markdown

---

title: "Response to Reviewers 2"

author: "S. O'Neill"

format: pdf

editor: source

---

Dear editor and reviewers

Thank you very much for a second review of our manuscript, which has further improved it, in our view.

# Review comments to the author

## Reviewer #1

We thank the reviewer for their efforts and for recommending our manuscript for publication.

## Reviewer #2

We thank the reviewer for additional comments, which we address below:

**- To increase the accuracy of the expression, the paragraph**

We have amended the first paragraph of _Discussion/Force measurement_ as suggested by reviewer #2. It now reads:

> The force measurement platform was programmed to read sensor data continuously rather

than enforcing a fixed frequency of sampling, which resulted in a small degree of variation

in sampling rates. This is unlikely to make any difference in the current results, but for

future implementation it could be useful to **upgrade the acquisition software in order to

synchronize sampling with an accurate time base generated by a hardware timer (present inside

the ATmega328 processor).**

Text highlighted in bold, has been revised.

**- Before equation 2 it could be useful to change the**

We have amended the relevant sentence in _Methods and materials/Pressure probe test setup/Time series data analyses_ as suggested by reviewer #2. It now reads:

> Equation 2 **describes the way to obtain the optimal warping path by minimization of the the Euclidean distance.**

Text highlighted in bold, has been revised.

**- In Discussion section at "Rate of pressure application" paragraph it could be useful to add a comment about

the results presented in Figure 13 and Table 3 indicating that the operator needs sufficient instruction to actually understand the task at hand.**

We agree and thank the reviewer for pointing this out. That point was, to some extent, made in the preceding section, but we agree it bears being made explicit in the "Rate of pressure application" section also. We have added the following to that section:

> **The small number of testers who, despite an audio cue, had a rate of application twice that of the target rate, 

illustrates the need for sufficient instruction of testers.**

> We recommend the use of a sound cue to ensure that testers can reliably maintain the correct rate of pressure application with little variation **and we recommend that instructions in the use of pressure probes, be supplemented by practice sessions supervised by an experienced tester.**

Text highlighted in bold, has been added

We thank the reviewer for recommending to accept our manuscript after these minor revisions.

On behalf of the authors

S. O'Neill -- Sep 20th 2023

---

## [Decision Letter · Decision Letter 2]

29 Sep 2023

Validation of a spring loaded probe for single and repeat pressure pain testing, including public domain specifications for design and manufacture

PONE-D-23-13550R2

Dear Dr. Søren O’Neill,

We’re pleased to inform you that your manuscript has been judged scientifically suitable for publication and will be formally accepted for publication once it meets all outstanding technical requirements.

Kind regards,

Livia Petrescu

Academic Editor

PLOS ONE

Additional Editor Comments (optional):

Reviewers' comments:

Reviewer's Responses to Questions

**Comments to the Author**

1. If the authors have adequately addressed your comments raised in a previous round of review and you feel that this manuscript is now acceptable for publication, you may indicate that here to bypass the “Comments to the Author” section, enter your conflict of interest statement in the “Confidential to Editor” section, and submit your "Accept" recommendation.

Reviewer #2: All comments have been addressed

2. Is the manuscript technically sound, and do the data support the conclusions?

Reviewer #2: Yes

3. Has the statistical analysis been performed appropriately and rigorously? 

Reviewer #2: Yes

4. Have the authors made all data underlying the findings in their manuscript fully available?

Reviewer #2: Yes

5. Is the manuscript presented in an intelligible fashion and written in standard English?

Reviewer #2: Yes

6. Review Comments to the Author

Reviewer #2: (No Response)

7. PLOS authors have the option to publish the peer review history of their article (what does this mean?). If published, this will include your full peer review and any attached files.

Reviewer #2: No

---

## [Editor Report · Acceptance letter]

3 Oct 2023

PONE-D-23-13550R2 

Validation of a spring loaded probe for single and repeat pressure pain testing, including public domain specifications for design and manufacture 

Dear Dr. O'Neill:

I'm pleased to inform you that your manuscript has been deemed suitable for publication in PLOS ONE. Congratulations! Your manuscript is now with our production department. 

Kind regards, 

on behalf of

Dr. Livia Petrescu 

Academic Editor

PLOS ONE